# Off-Policy Evaluation of Slate Bandit Policies via Optimizing Abstraction

## ABSTRACT

We study *off-policy evaluation* (OPE) in the slate contextual bandits where a policy selects multi-dimensional actions known as slates. This problem is widespread in recommender systems, search engines, marketing, to medical applications, however, the typical Inverse Propensity Scoring (IPS) estimator suffers from substantial variance due to large action spaces, making effective OPE a significant challenge. The PseudoInverse (PI) estimator has been introduced to mitigate variance by assuming linearity in the reward function, but this can result in significant bias as this assumption is hard-to-verify from observed data and is often substantially violated. To address the limitations of previous estimators, we develop a novel estimator for OPE of slate bandits, called *Latent IPS* (LIPS), which defines importance weights in a low-dimensional slate abstraction space where we optimize slate abstractions to minimize the bias and variance of LIPS in a data-driven way. By doing so, LIPS can substantially reduce the variance of IPS without imposing restrictive assumptions on the reward function structure like linearity. Through empirical evaluation, we demonstrate that LIPS substantially outperforms existing estimators, particularly in scenarios with non-linear rewards and large slate spaces.

## ACM Reference Format:
Anonymous Author(s). 2024. Off-Policy Evaluation of Slate Bandit Policies via Optimizing Abstraction. In *Woodstock '18: ACM Symposium on Neural Gaze Detection, June 03–05, 2018, Woodstock, NY*. ACM, New York, NY, USA, 19 pages. https://doi.org/10.1145/nnnnnnn.nnnnnnn

## 1 INTRODUCTION

Slate bandits play a pivotal role in many online services, such as recommender and advertising systems. In these services, a decision-making algorithm or *policy* selects a combinatorial and potentially high-dimensional slate composed of multiple sub-actions. For instance, visual advertisements consist of various components such as title, key visual, and background image. Each of these components significantly influences user interests and revenue outcomes. Another relevant application is in medical treatment, where the aim is to find the optimal combination of doses to improve medical outcomes. Although these systems typically have an abundance of logged data, accurately estimating the performance of a counterfactual policy through Off-Policy Evaluation (OPE) can be exceedingly challenging. This challenge arises largely due to the exponential

variance associated with slate-wise importance weighting, particularly when dealing with combinatorial actions or slate spaces [36].

Despite the challenges in OPE with slate structures, it remains a crucial task for safely evaluating and improving the effectiveness of real-world interactive systems [26]. Therefore, several *PseudoInverse* (PI) estimators have been proposed [33, 36, 41]. These estimators use slot-wise importance weights to reduce the variance of IPS and have been shown to enable unbiased OPE under the *linearity* assumption on the reward. This assumption requires that the expected reward should be linearly decomposable, ignoring interaction effects among different slots. While PI often outperforms IPS under linear reward structures, it can still produce significant bias when linearity does not hold. In addition, the variance of PI remains extremely high if there are many unique sub-actions [28]. It is worth noting that while several estimators have been developed for OPE in ranking action spaces [16–18, 21], they are not applicable to evaluate slate bandits. This is because these estimators assume that the rewards are available for every position in a ranking, whereas in our setup, only slate-wise rewards are available.

To overcome the limitations of slate OPE, we propose a novel approach called *Latent IPS* (LIPS), which redefines importance weights in a low-dimensional slate abstraction space. LIPS is inspired by recent advances in OPE for large action spaces where the marginalized IPS (MIPS) estimator leverages pre-defined action embeddings to provably reduce variance [28, 29], but our estimator does not assume that action embeddings are already observed in the logged data a priori as MIPS. Through theoretical analysis, we show that LIPS can substantially reduce variance compared to IPS while also being unbiased if the slate abstraction is *sufficient* in that it retains sufficient information to characterize the reward function. In addition, LIPS can achieve lower bias and variance than PI under appropriate slate abstractions since our sufficiency condition is more relaxed than the linearity assumption and the variance of LIPS only depends on the size of the abstraction space, which can be more compact than the sub-action spaces. Interestingly, our analysis on the bias and variance of LIPS also implies that its mean-squared-error (MSE) may be minimized when strategically using an insufficient slate abstraction, potentially resulting in even greater variance reduction while being nearly unbiased. Based on this theoretical analysis, we also develop a procedure to optimize slate abstraction to directly minimize the bias and variance of LIPS. This is a particular distinction of LIPS from MIPS [28] and its extension [29], which assume that useful action embeddings already exist. Empirical results on extreme classification datasets (which can be transformed into bandit feedback with slate actions) demonstrate that LIPS enables more accurate slate OPE than IPS, PI, a naive extension of MIPS, and their doubly-robust variants on a variety of non-linear reward functions and large slate action spaces.

Our contributions can be summarized below.

- We propose the LIPS estimator, which leverages an abstraction of slates to substantially improve OPE of slate bandits.
- We develop a fully data-driven procedure to optimize an abstraction function to directly minimize the MSE of LIPS.
- We empirically demonstrate that LIPS with an optimized abstraction outperforms existing estimators (such as PI, IPS, and MIPS) for a range of scenarios with non-linear rewards.

## 2 OFF-POLICY EVALUATION FOR SLATES

This work considers a *slate* contextual bandit problem where $x \in \mathcal{X} \subseteq \mathbb{R}^d$ represents a context vector (e.g., user demographics, consumption history, weather) and $s \in \mathcal{S} := \prod_{l=1}^{L} \mathcal{A}_l$ denotes a slate action. A slate consists of several sub-actions, i.e., $s = (a_1, a_2, \ldots, a_L)$ where each sub-action $a_l$ is chosen from the corresponding action set $\mathcal{A}_l$, which may differ across different slots. For instance, in an email campaign, $\mathcal{A}_1$ may be a set of subject lines, while $\mathcal{A}_2$ may represent whether or not to include visuals in a promotion email.

We refer to a function $\pi : \mathcal{X} \to \Delta(\mathcal{S})$ as a slate bandit policy, which maps each context to a distribution over the slate space. In particular, throughout the paper, we will focus on a *factored* policy, i.e., $\pi(s \mid x) = \prod_{l=1}^{L} \pi(a_l \mid x)$ for brevity of exposition. Let then $r$ be a reward associated with a slate $s$, which is considered sampled from an unknown conditional distribution $p(r \mid x, s)$. We are interested in OPE in this slate bandit setup, where we are given a logged dataset $\mathcal{D} := \{(x_i, s_i, r_i)\}_{i=1}^{n}$ collected by a logging policy $\pi_0$ where $(x, s, r) \sim p(x)\pi_0(s|x)p(r|x, s)$. In particular, we aim to estimate the following expected performance of a new policy $\pi$ (which is called a target policy and is often different from $\pi_0$):

$$V(\pi) := \mathbb{E}_{(x,s)\sim p(x)\pi(s \mid x)}[q(x, s)],$$

where $q(x, s) := \mathbb{E}[r \mid x, s]$ is the expected reward function given context $x$ and slate $s$. In particular, our goal is to develop an estimator $\hat{V}$ capable of accurately estimating the performance of $\pi$ relying only on the logged data $\mathcal{D}$. The accuracy of an estimator is measured by the mean-squared-error (MSE):

$$\text{MSE}(\hat{V}(\pi)) : = \mathbb{E}_{\mathcal{D}}\left[\left(V(\pi) - \hat{V}(\pi; \mathcal{D})\right)^2\right]$$
$$= \text{Bias}(\hat{V}(\pi; \mathcal{D}))^2 + \mathbb{V}_{\mathcal{D}}\left[\hat{V}(\pi; \mathcal{D})\right],$$

where the bias and variance of $\hat{V}$ are defined respectively as

$$\text{Bias}(\hat{V}(\pi; \mathcal{D})) := \mathbb{E}_{\mathcal{D}}[\hat{V}(\pi; \mathcal{D})] - V(\pi),$$
$$\mathbb{V}_{\mathcal{D}}\left[\hat{V}(\pi; \mathcal{D})\right] := \mathbb{E}_{\mathcal{D}}\left[(\hat{V}(\pi; \mathcal{D}) - \mathbb{E}_{\mathcal{D}}[\hat{V}(\pi; \mathcal{D})])^2\right].$$

**Existing estimators.** We now summarize key existing estimators and their limitations.

*Inverse Propensity Scoring (IPS) [32].* IPS reweighs the observed rewards by the ratios of slate probabilities under the target and logging policies (slate-wise importance weight) as

$$\hat{V}_{\text{IPS}}(\pi; \mathcal{D}) := \frac{1}{n} \sum_{i=1}^{n} \left(\prod_{l=1}^{L} \frac{\pi(a_{i,l} \mid x_i)}{\pi_0(a_{i,l} \mid x_i)}\right) r_i = \frac{1}{n} \sum_{i=1}^{n} w(x_i, s_i)r_i, \quad (1)$$

where $w(x, s) := \pi(s|x)/\pi_0(s|x)$ is the slate-wise importance weight. IPS is unbiased under some identification assumptions such as common support (i.e., $\pi(s \mid x) > 0 \to \pi_0(s \mid x) > 0, \forall(x, s)$). However, its critical issue is its exponential variance, which arises due to the potentially astronomical size of slate action spaces [36].

*PseudoInverse (PI) [36].* To deal with the exponential variance of IPS, the PI estimator leverages only the slot-wise importance weights (compared to slate-wise importance weighting of IPS) as

$$\hat{V}_{\text{PI}}(\pi; \mathcal{D}) := \frac{1}{n} \sum_{i=1}^{n} \left(\sum_{l=1}^{L} \frac{\pi(a_{i,l} \mid x_i)}{\pi_0(a_{i,l} \mid x_i)} - L + 1\right) r_i. \quad (2)$$

Since PI relies only on slot-wise importance weights, its variance only scales with the number of unique sub-actions (i.e., $\sum_{l=1}^{L} |\mathcal{A}_l|$). Thus, the variance of PI is often smaller than that of IPS whose weight scales with the cardinality of the slate space (i.e., $\prod_{l=1}^{L} |\mathcal{A}_l|$). PI has also been shown to enable an unbiased performance evaluation, i.e $\mathbb{E}_{\mathcal{D}}[\hat{V}_{\text{PI}}(\pi; \mathcal{D})] = V(\pi)$, under the linearity assumption, which requires that the reward function is linearly decomposable and there exists some (latent) intrinsic reward functions $\{\phi_l\}_{l=1}^{L}$ such that $q(x, s) = \sum_{l=1}^{L} \phi_l(x, a_l)$ for every context $x$ and slot $l$. In other words, this assumption essentially ignores every possible interactions across different slots.

Although PI improves the MSE over IPS under linear reward functions, its bias is no longer controllable when linearity does not hold, which is often the case in highly non-linear real-world environments. In addition, PI may still suffer from extremely high variance when $\sum_{l=1}^{L} |\mathcal{A}_l|$ is large (i.e., when there are many unique sub-actions). These limitations of PI and IPS motivate the development of a new estimator for slate OPE that can substantially reduce the variance while being (nearly) unbiased without any unrealistic assumptions on the reward function structure.

## 3 THE LIPS ESTIMATOR

To deal with the issues of IPS and PI, we now propose the LIPS estimator that enables more effective OPE by leveraging *slate abstraction* rather than positing restrictive assumptions on the reward. At a high level, LIPS defines importance weights in a (low-dimensional) latent slate space $\mathcal{Z}$, which can either be discrete or continuous[1] and is induced via a slate abstraction function $\phi_\theta : \mathcal{S} \to \mathcal{Z}$, parametrized by $\theta$. Specifically, our LIPS estimator is defined as

$$\hat{V}_{\text{LIPS}}(\pi; \mathcal{D}) := \frac{1}{n} \sum_{i=1}^{n} \underbrace{\frac{\pi(\phi_\theta(s_i) \mid x_i)}{\pi_0(\phi_\theta(s_i) \mid x_i)}}_{:= w(x_i, \phi_\theta(s_i))} r_i \quad (3)$$

where $\pi(z|x) := \sum_{s \in \{s' \in \mathcal{S} \mid \phi_\theta(s')=z\}} \pi(s|x)$ is a marginal distribution of an abstracted slate induced by policy $\pi$, and $w(x, z) := \pi(z|x)/\pi_0(z|x)$ is the *latent importance weight*. Note that we can readily extend LIPS to the case with a context-dependent and stochastic abstraction based on a parameterized distribution $p_\theta : \mathcal{X} \times \mathcal{S} \to \Delta(\mathcal{Z})$. This means that we can generalize Eq. (3) as

$$\hat{V}_{\text{LIPS}}(\pi; \mathcal{D}) := \frac{1}{n} \sum_{i=1}^{n} \frac{p_\theta(z_i \mid x_i; \pi)}{p_\theta(z_i \mid x_i; \pi_0)} r_i$$

where $z_i \sim p_\theta(\cdot \mid x_i, s_i)$ and $p_\theta(z \mid x; \pi) := \sum_{s \in \mathcal{S}} \pi(s \mid x)p_\theta(z \mid x, s)$ $= \sum_{s \in \mathcal{S}} p_\theta(s, z \mid x; \pi)$. This extension allows for a flexible control of the bias-variance tradeoff of LIPS and ensures a tractable optimization of abstraction. The central idea of LIPS is to circumvent

---

[1]In the remainder of the paper, we rely on a discrete abstraction for ease of exposition, but a continuous abstraction space can also be considered under stochastic abstraction.

the reliance on slate- or slot-wise importance weights, substantially improving the variance from IPS and PI while avoiding the linear-reward assumption like PI. The following formally analyzes LIPS and shows its statistical advantages over existing estimators. We also develop a data-driven optimization procedure for slate abstractions to directly minimize the MSE of LIPS.

## 3.1 Theoretical Analysis

First, we analyze the bias of LIPS based on the notion of *sufficient* slate abstraction (but we will later show that intentionally using an insufficient slate abstraction is indeed a better implementation and present how to data-drivenly obtain a better slate abstraction.)

*Definition 3.1.* (Sufficient Slate Abstraction) A slate abstraction function $\phi_\theta$ is said to be sufficient if it satisfies $q(x, s) = q(x, s')$ for all $x \in \mathcal{X}$, $s \in \mathcal{S}$, and $s' \in \{s'' \in \mathcal{S} \mid \phi_\theta(s) = \phi_\theta(s'')\}$.

An abstraction function is *sufficient* if it aggregates only slates that have the same expected reward, and it means that the latent slate space $\mathcal{Z}$ retains sufficient information to characterize the reward function. For example, identity abstraction $\phi(s) = s$ is always sufficient (LIPS is reduced to IPS in this case). Note here that this notion of sufficiency does not impose any particular restriction on the reward function form such as linearity. Furthermore, sufficient slate abstractions may not be unique, and there could potentially be many sufficient slate abstractions.

The following demonstrates that LIPS is unbiased when given a sufficient abstraction function. We also characterize the bias of LIPS when we use a stochastic abstraction, which may not be sufficient.

THEOREM 3.2. *(Unbiasedness of LIPS) LIPS is unbiased, i.e.,*

$$\mathbb{E}_{\mathcal{D}}[\hat{V}_{\text{LIPS}}(\pi; \mathcal{D})] = V(\pi),$$

*if a given slate abstraction function $\phi_\theta$ is sufficient. See Appendix B.3 for the proof.*

THEOREM 3.3. *(Bias of LIPS) The bias of LIPS given a stochastic slate abstraction $p_\theta$ is*

$$\text{Bias}(\hat{V}_{\text{LIPS}}(\pi; \mathcal{D})) \tag{4}$$

$$= \mathbb{E}_{p(x)p_\theta(z|x;\pi_0)} \Big[ \sum_{j < k \leq |\mathcal{S}|} p_\theta(s_j \mid x, z; \pi_0) p_\theta(s_k \mid x, z; \pi_0) \tag{5}$$

$$\times (q(x, s_j) - q(x, s_k)) \times (w(x, s_k) - w(x, s_j)) \Big],$$

*where $p_\theta(s \mid x, z; \pi) = p_\theta(s, z \mid x; \pi)/\pi(z \mid x)$. See Appendix B.4 for the proof.*

In particular, Theorem 3.3 implies that the bias of LIPS is characterized by the following factors.

(1) identifiability of the slates from their abstractions:
$p_\theta(s_j|x, z; \pi_0) p_\theta(s_k|x, z; \pi_0)$

(2) difference in the expected rewards between a pair of slates:
$q(x, s_j) - q(x, s_k)$

(3) difference in the slate-wise importance weights between a pair of slates: $w(x, s_k) - w(x, s_j)$

When slate $s$ is near-deterministic given $(x, z)$, $p_\theta(s_j|x, z; \pi_0)$ approaches either zero or one, making $p_\theta(s_j|x, z; \pi_0) p_\theta(s_k|x, z; \pi_0)$ (the first factor) close to zero. This suggests that if the latent slate space contains sufficient information to reconstruct the original

slates, the bias of LIPS remains small even with a stochastic slate abstraction. Moreover, the second factor indicates how predictive the rewards are with latent variable $z$. In particular, when the latent variable is *nearly sufficient* in the sense that the slates that have similar expected rewards have similar abstraction distribution, $q(x, s_j) - q(x, s_k)$ becomes small when $p_\theta(s_j|x, z; \pi_0) p_\theta(s_k|x, z; \pi_0)$ is large, leading to a reduced bias of LIPS. Thus, the analysis implies that the bias of LIPS remains small if the latent slate space is finer-grained and abstraction is closer to deterministic, which makes the latent variable $z$ more predictive of slate $s$ and reward $r$.

Next, we analyze the variance reduction of LIPS against IPS, which can be substantially large depending on the coarseness and stochasticity of the latent slate space.

THEOREM 3.4. *(Variance Reduction) Given a sufficient slate abstraction function $\phi_\theta$, we have*

$$n\big(\mathbb{V}_{\mathcal{D}}[\hat{V}_{\text{IPS}}(\pi; \mathcal{D})] - \mathbb{V}_{\mathcal{D}}[\hat{V}_{\text{LIPS}}(\pi; \mathcal{D})]\big)$$
$$= \mathbb{E}_{p(x)\pi_0(\phi_\theta(s)|x)} \Big[ \mathbb{E}_{p(r|x,\phi_\theta(s))}[r^2] \, \mathbb{V}_{\pi_0(s|x,\phi_\theta(s))} [w(x, s)] \Big]. \tag{6}$$

*See Appendix B.5 for the proof.*

There are two key factors that characterize the extent of variance reduction. The first factor is the second moment of the reward, which becomes large when the reward is noisy. The second factor is the variance of the slate-wise importance weight $w(x, s)$ with respect to the conditional distribution $\pi_0(s|x, \phi(s)) (= \pi_0(s|x)/\pi_0(\phi(s)|x))$, which becomes large when **(i)** $w(x, s)$ takes large values and **(ii)** $\pi_0(s|x, \phi(s))$ remains adequately stochastic. This implies that the variance reduction of LIPS can be exponential as the variance of the slate importance weight can grow exponentially with the slate size $L$. Moreover, we can control the extent of variance reduction via the cardinality of the latent slate space and entropy of slate abstraction, i.e., a coarser and more stochastic abstraction reduces variance more. This theoretical observation indeed implies that a sufficient abstraction may not minimize the MSE of LIPS. Table 1 provides a toy example illustrating a situation where LIPS with an insufficient abstraction can indeed achieve a lower MSE than that with a sufficient abstraction. Specifically, LIPS with a sufficient abstraction achieves the MSE of 1.0 with zero bias as per Theorem 3.2. However, a lower MSE can be realized by intentionally using an insufficient abstraction. This is because we can gain a large variance reduction (0.2 - 1.0 = -0.8) by allowing only a small squared bias (+0.25), and hence using a sufficient abstraction does not result in the optimal MSE of LIPS. Therefore, instead of discussing how to find a sufficient abstraction, the following describes a data-driven approach to *optimize* it in a way that minimizes the MSE of LIPS.

## 3.2 Optimizing Slate Abstractions

The analysis from the previous section indicates that **the bias-variance tradeoff of LIPS is determined by the granularity of the latent slate space $\mathcal{Z}$ and stochasticity of slate abstraction distribution $p_\theta(z \mid x, s)$**, which also suggests that the MSE of LIPS might be minimized with an *insufficient* abstraction that yields even greater variance reduction while producing only some small bias. This insight naturally encourages us to directly minimize the bias and variance of LIPS when optimizing slate abstraction rather than

**Table 1: A toy example illustrating the potential advantage of strategic variance reduction with an insufficient abstraction. LIPS with an insufficient (but optimized) abstraction produces much smaller variance while introducing some small bias, resulting in a smaller MSE than LIPS with a sufficient abstraction.**

| | bias | variance | **MSE** (= bias$^2$ + variance) |
|---|---|---|---|
| LIPS with a sufficient abstraction | 0.0 | 1.0 | **1.00** $\left( = (0.0)^2 + 1.0 \right)$ |
| LIPS with an insufficient (but optimized) abstraction | 0.5 | 0.2 | **0.45** $\left( = (0.5)^2 + 0.2 \right)$ |

myopically searching for a sufficient abstraction. Specifically, we propose to optimize slate abstraction distribution $p_\theta$ via

$$(\hat{\theta}, \hat{\psi}, \hat{\omega}) = \arg\max_{\theta, \psi} \min_{\omega} \sum_{i=1}^{n} \mathcal{L}(x_i, s_i, \pi_0; \theta, \psi, \omega) \quad (7)$$

where

$$\mathcal{L}(x, s, \pi_0; \theta, \psi, \omega) = \underbrace{\mathbb{E}_{p_\theta(z \mid x, s; \pi_0)} \left[ \log p_\psi(s \mid x, z; \pi_0) \right]}_{\text{bias reduction: } p_\theta(s_j \mid x, z; \pi_0) p_\theta(s_k \mid x, z; \pi_0)} \quad (8)$$

$$+ \underbrace{\mathbb{E}_{p_\theta(z \mid x, s; \pi_0)} \left[ (r - \hat{q}_\omega(x, z))^2 \right]}_{\text{bias reduction: } q(x, s_j) - q(x, s_k)}$$

$$- \underbrace{\beta \mathrm{KL}(p_\theta(z \mid x, s; \pi_0) \,||\, p_\psi(z \mid x; \pi_0))}_{\text{variance reduction: } p_\theta(s \mid x, z; \pi_0)}. \quad (9)$$

$\theta$, $\psi$, and $\omega$ are the parameters of slate abstraction $p_\theta(z|x, s; \pi_0)$, slate reconstruction $p_\psi(s|x, z; \pi_0)$, and reward construction $\hat{q}_\omega(x, z)$ models.[2] The first two terms of the loss function aim for bias reduction, while the last term controls variance reduction of LIPS. More specifically, the first term measures the identifiability of the slates and the second term measures how predictive the rewards are based on the latent variable. In contrast, the last term works as a regularization to control the stochasticity of $p_\theta(s|x, z; \pi_0)$ by making $p_\theta(z \mid x, s; \pi_0)$ closer to $p_\psi(z \mid x; \pi_0)$. This will indeed regularize the latent importance weights and make them close to 1 everywhere (as described in the appendix in detail), and thus we can expect a larger variance reduction of LIPS when the last term is dominant. The regularization weight $\beta$ is thus considered the key hyper-parameter that governs the bias-variance tradeoff of LIPS (i.e., a smaller $\beta$ implies a smaller bias and larger variance of LIPS, while a larger $\beta$ leads to a larger bias and smaller variance). This hyper-parameter can be tuned by existing parameter tuning methods such as SLOPE [34] and PAS-IF [39], which are feasible using only observed logged data $\mathcal{D}$. Section 5 empirically investigates how LIPS works with these existing parameter tuning methods.

## 4 RELATED WORK

***Off-Policy Evaluation.*** Off-Policy Evaluation (OPE) has gained increasing attention in fields ranging from recommender systems to personalized medicine as a safe alternative to online A/B tests, which might be risky, slow, and sometimes even unethical [15, 27]. Among many OPE estimators studied in the single-action setting, DM and IPS [32] are commonly considered baseline estimators. DM

trains a reward prediction model to estimate the policy value. Although DM does not produce large variance, it can be highly biased when the reward predictor is inaccurate. In contrast, IPS allows for unbiased estimation under standard identification assumptions, but it often suffers from high variance due to large importance weights. Doubly Robust (DR) [9] is a hybrid method that combines DM and IPS to improve variance while remaining unbiased. However, its variance can still be very high under large action spaces [28]. As a result, the primary objective of OPE research has been to effectively balance the bias and variance, and numerous estimators have been proposed to address this statistical challenge [22, 33, 42].

In comparison, the slate contextual bandit setting has been relatively under-explored despite its prevalence in real practice [8, 13, 36] and the necessity for significant variance reduction due to combinatorial action spaces. Existing approaches, such as PI [36] and its variants [33, 41], strongly rely on the linearity assumption of the reward function. However, when this assumption does not hold, PIs are no longer unbiased. Moreover, their variance can be substantial when there are many unique sub-actions. Compared to PIs, LIPS improves the variance without making restrictive assumptions about the reward function form via leveraging slate abstraction. LIPS can have a much lower bias and variance than PI when slate abstraction is appropriately optimized. Our approach is also relevant to the MIPS [28] and OffCEM [29] estimators, which employ action embeddings in the single-action setting. Although they assume that useful action embeddings are already present in the logged data, we develop a novel method to optimize slate abstractions based on the logged bandit data to directly improve the resulting estimator. The next section empirically demonstrates that our data-driven optimization procedure to obtain an appropriate slate abstraction is a crucial component of LIPS to outperform existing estimators (IPS and PI) as well as MIPS [28].

It is important to note that there exists a relevant line of research called OPE of ranking policies, where the action space consists of ordered sets of items [16–18, 21]. While this ranking setting closely resembles that of the slate, all existing estimators require the observation of slot-wise rewards, which makes them inapplicable to our slate setup, where only the reward for each slate is observed (thus the slate OPE problem is fundamentally more difficult). However, the idea of abstraction could be effective in ranking OPE as well, and we consider this to be a valuable future direction.

***Abstraction in Bandits.*** Action abstraction is often used in bandits to accelerate policy learning through efficient exploration [20, 23, 30, 31]. For instance, [20, 31] introduce a tree-based hierarchical structure in the action space, where each node of the tree can be seen as an action abstraction that aggregates similar actions in its child nodes. In this way, these methods reduce the number of

---

[2]The pseudo-code (Algorithm 1) of this abstraction optimization procedure can be found in the appendix. Appendix B provides a more extensive bias-variance analysis of LIPS when given a stochastic abstraction $p_\theta$.

effective actions during exploration, thereby improving sample efficiency in online policy learning. A similar approach has also been studied for enabling more efficient learning of ranking policies [30]. While these ideas serve as inspiration of our approach, their focus is policy learning, and no existing work considers leveraging action abstraction to improve OPE for slate bandits.

***Representation Learning for Causal Inference and Recommendations.*** Latent variable modeling has been deemed effective in causal inference when identifying confounders from proximal variables [19] or recovering confounders and treatments from high-dimensional data such as texts [12, 40, 44]. Among them, the closest to ours is Veitch et al. [40], which takes into account reward prediction and treatment reconstruction loss when optimizing latent text representations to perform causal inference regarding text data. However, these methods have been developed for the conventional task of estimating the average treatment effect, and we are not aware of any similar approaches or loss functions in OPE.

In another line of research, latent action representations are often learned to exploit the structure in the action space in reinforcement learning [3, 6, 7, 46, 47]. In particular, Deffayet et al. [7] use a Variational AutoEncoder (VAE) model to pretrain latent slate space from logged data to improve recommendations. Consequently, they achieve a better exploration-exploitation tradeoff regarding long-term objectives and item diversity through VAE. While the VAE model of [7] is somewhat similar to our optimization procedure for slate abstraction described in Section 3.2, our loss function is derived from the theoretical analysis of LIPS and aims to directly improve the MSE in slate OPE rather than recommendation effectiveness.

## 5 EMPIRICAL EVALUATION

This section empirically compares LIPS with many relevant estimators on two real-world datasets, namely Wiki10-31K and Eurlex-4K provided in the Extreme Classification repository [5]. Our code is available in an anonymous Google Drive folder and will be made public on a GitHub repository upon publication.

### 5.1 Experiment Setup

We follow the standard "supervised-to-bandit" procedure to conduct an OPE experiment based on classification data, as used in many previous studies [11, 33, 35, 42]. Specifically, we use the extreme classification datasets called Wiki10-31K and Eurlex-4K. These datasets consist of many documents associated with a large number of labels, which are approximately 31K for Wiki10-31K and 4K for Eurlex-4K. More detailed dataset statistics are provided in Appendix A.

To simulate a slate bandit scenario, we regard the documents as contexts ($\mathbf{x}$). Wiki10-31K and Eurlex-4K inherently contain some text data that represent the documents, and we apply Sentence-Transformer [25] and PCA [1] to encode the raw texts into 20-dimensional contexts for both these datasets. Next, to define the slate action space, we first extract the top 1,000 dense labels in terms of the number of positive documents. Then, we randomly sample $L \times 10$ labels to form $L$ disjoint action sets $\{\mathcal{A}_l\}_{l=1}^{L}$ corresponding to $L$ distinct slots, with each set having a size of $|\mathcal{A}_l| = 10$. For each action $a \in \mathcal{A}_l$, we first define $q_l(\mathbf{x}, a) = 1 - \eta_a$ if the action has a positive label and $q_l(\mathbf{x}, a) = \eta_a$ otherwise, where $\eta_a$ is a noise

parameter sampled separately for each action from a uniform distribution with range $[0, 0.5]$. Given that we never know the nature of real-world reward functions within a slate, we simulate various *non-linear* relationships (where linearity of PI is violated) that bring complex interactions across different slots. Specifically, we use the following synthetic reward functions.

- reward model (1)

$$q(\mathbf{x}, \mathbf{s}) = \frac{1}{\lfloor L/2 \rfloor} \sum_{l=1}^{\lfloor L/2 \rfloor} q_l(\mathbf{x}, a_l) + \frac{1}{\lfloor L/2 \rfloor - 1} \sum_{l=1}^{\lfloor L/2 \rfloor - 1} w(a_l, a_{l+1})$$

- reward model (2)

$$q(\mathbf{x}, \mathbf{s}) = \frac{1}{\lfloor L/2 \rfloor} \Big( q_1(\mathbf{x}, a_1) + \sum_{l=2}^{\lfloor L/2 \rfloor} w(a_{l-1}, a_l) q_l(\mathbf{x}, a_l) \Big)$$

- reward model (3)

$$q(\mathbf{x}, \mathbf{s}) = \frac{1}{2} \Big( \min_{l=1}^{\lfloor L/2 \rfloor} q_l(\mathbf{x}, a_l) + \max_{l=1}^{\lfloor L/2 \rfloor} q_l(\mathbf{x}, a_l) \Big)$$

where $\lfloor c \rfloor := \max\{n \in \mathbb{Z} \mid n \le c\}$ is the floor function. $w(a_l, a_{l+1})$ is a scalar value to represent the co-occurrence effect between $a_l$ and $a_{l+1}$ and it is sampled from the standard normal distribution. Note that, when defining these reward functions, we use only 50% of the slots $l = 1, 2, \ldots, \lfloor L/2 \rfloor$ among total of $L$ slots. By doing so, we can make $(a_1, a_2, \cdots, a_{\lfloor L/2 \rfloor})$ a sufficient abstraction, which enables us to study if LIPS achieves more accurate estimation (lower MSE) by intentionally using an insufficient abstraction obtained by the optimization procedure from Section 3.2.

Given these reward functions, we sample reward $r$ from a normal distribution as $r \sim \mathcal{N}(q(\mathbf{x}, \mathbf{s}), \sigma^2)$ with a standard deviation of $\sigma = 0.1$. To obtain logging ($\pi_0$) and evaluation ($\pi$) policies, we first train a base classifier $\tilde{q}(\mathbf{x}, \mathbf{s})$ via the REINFORCE method [43] and then define the policies as follows:

$$\pi_0(\mathbf{s} \mid \mathbf{x}) = \prod_{l=1}^{L} \Big( (1 - \epsilon_0) \frac{\exp(\gamma_0 \cdot \tilde{q}(\mathbf{x}, a_l))}{\sum_{a \in \mathcal{A}_l} \exp(\gamma_0 \cdot \tilde{q}(\mathbf{x}, a))} + \frac{\epsilon_0}{|\mathcal{A}_l|} \Big),$$

$$\pi(\mathbf{s} \mid \mathbf{x}) = \prod_{l=1}^{L} \Big( (1 - \epsilon) \mathbb{I}\{a_l = a_l^*\} + \frac{\epsilon}{|\mathcal{A}_l|} \Big),$$

where $a_l^* := \arg\max_{a \in \mathcal{A}_l} \tilde{q}(\mathbf{x}, a)$. $\gamma \in \mathbb{R}$ and $\epsilon_0, \epsilon \in [0, 1]$ are the experiment parameters that control the stochasticity of $\pi_0$ and $\pi$. We use $(\gamma, \epsilon_0, \epsilon) = (-1.0, 0.1, 0.3)$ in the main text.

***Compared estimators.*** We compare LIPS with Direct Method (DM) [4], IPS [32], PI [36], and MIPS [28]. DM is a regression-based estimator that estimates the policy value based on an estimated reward function $\hat{q}(\mathbf{x}, \mathbf{s})$, which is learned by a neural network in our experiment. For LIPS, we employ a discrete abstraction whose dimension is 100 ($|\mathcal{Z}| = 100$). When optimizing a slate abstraction, we select the hyper-parameter $\beta$ from $\{0.01, 0.1, 1.0, 10.0\}$ based only on the available logged data via the SLOPE algorithm [34, 38]. Appendix A provides some more details of SLOPE and describes how we parameterize and optimize the slate abstraction distribution $p_\theta$. Note that MIPS defines its importance weight taking only relevant slots (i.e., $l = 1, 2, \ldots, \lfloor L/2 \rfloor$) into account and is defined as $\hat{V}_{\text{MIPS}}(\pi; \mathcal{D}) := \frac{1}{n} \sum_{i=1}^{n} \big( \prod_{l=1}^{\lfloor L/2 \rfloor} \frac{\pi(a_{i,l} \mid \mathbf{x}_i)}{\pi_0(a_{i,l} \mid \mathbf{x}_i)} \big) r_i$, where the importance weights use only the first $\lfloor L/2 \rfloor$ slots as an action embedding

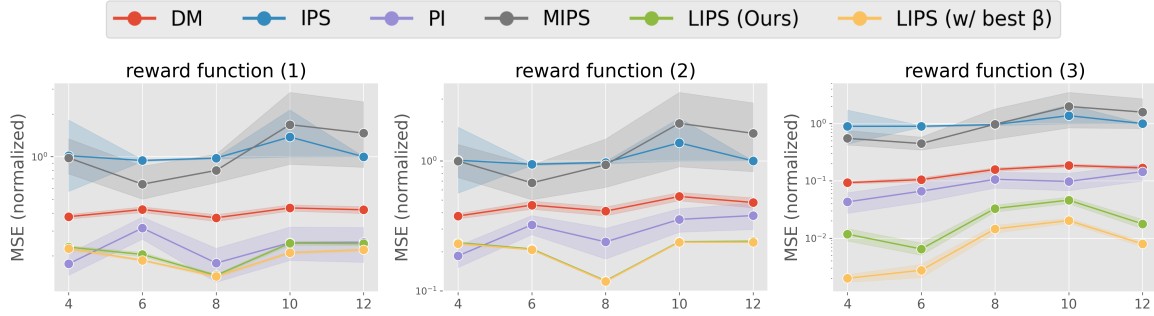

**Figure 1: Comparison of the estimators' MSE (normalized by the true value $V(\pi)$) with varying slate sizes ($L$) and with reward functions (1) - (3) on the Wiki10-31K dataset.**

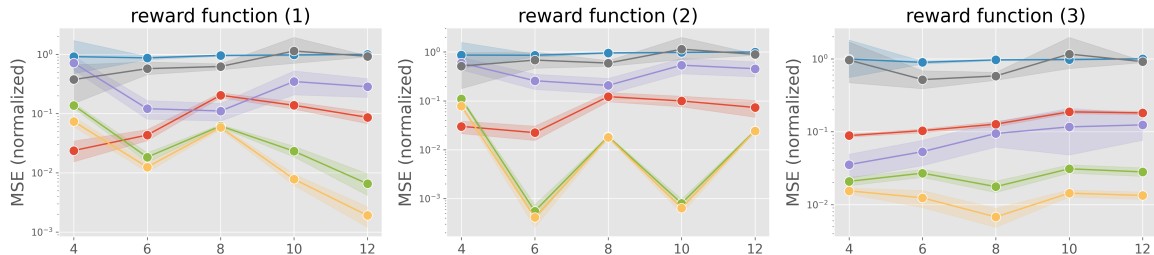

**Figure 2: Comparison of the estimators' MSE (normalized by the true value $V(\pi)$) with varying slate sizes ($L$) and with reward functions (1) - (3) on the Eurlex-4K dataset.**

leveraging the fact that the reward functions (1) - (3) depend only on these slots. MIPS is unbiased and has a lower variance than IPS, however, it is infeasible in practice since we do not know the true reward function. We include MIPS in our experiments since it is useful to investigate the effectiveness of intentionally using an insufficient abstraction.

In addition to these baselines, we also report the results of "LIPS (w/ best $\beta$)" as a reference. It indicates LIPS with the best value of $\beta$ selected based on the ground-truth MSE, which provides the best accuracy achievable by our LIPS framework with an oracle hyper-parameter selection.

## 5.2 Results and Discussion

The following reports and discusses the results obtained by running OPE simulations with 50 different logged datasets generated under different random seeds. We compare estimators' accuracy by their MSE normalized by the true policy value of the target policy, which is defined as $\text{MSE}(\hat{V}(\pi))/V(\pi)$. Note that we use $L = 8$, $|\mathcal{A}_l| = 10$ ($\forall l \in [L]$), and $n = 4000$ as default experiment parameters.

Figures 1 and 2 report estimators' MSE with varying slate sizes ($L$) and reward functions (1)-(3) on Wiki10-31K and Eurlex-4K, respectively. The results show that LIPS clearly outperforms existing estimators across a range of slate sizes ($L \in \{4, 6, \ldots, 12\}$) and various non-linear reward functions. In contrast, we find that PI is likely to produce higher MSEs with growing slate sizes $L$ on some reward functions. This is due to the fact that it produces larger variance when the slate size becomes larger. It is also true that the violation

of linearity is likely to produce a larger bias with larger slate sizes where the interaction effects between slots could be larger. DM often performs worse than LIPS due to its high bias that arises from the estimation error of its reward predictor $\hat{q}$. We also observe that LIPS substantially outperforms IPS and MIPS due to substantially reduced variance via latent importance weighting. The observation that LIPS consistently performs much better than MIPS is particularly intriguing since MIPS uses a sufficient abstraction based on the knowledge about the true reward functions. This observation empirically demonstrates that LIPS and its associated optimization procedure can strategically exploit an *insufficient* slate abstraction, leading to substantial variance reduction while introducing only a small amount of bias. Moreover, the comparison between LIPS and LIPS (w/ best $\beta$) implies that the feasible procedure of tuning the hyper-parameter $\beta$ via SLOPE [34] is often near-optimal even though it uses only observable data $\mathcal{D}$, while LIPS can still be slightly improved with a more refined tuning method.

We also obtain similar observations in Figures 3 and 4 where we compare estimators with varying data sizes ($n \in \{1000, 2000, \ldots, 16000\}$) on the Wiki10-31K and Eurlex-4K datasets. We observe that PI struggles particularly when the sample size is small due to its high variance and the linearity assumption. DM is highly biased on both datasets and performs even worse than PI in many cases because estimating the reward function in large slate spaces is particularly challenging. IPS and MIPS produce much larger variance than other methods due to slate-wise importance weighting, and we see that they do not converge even with the largest data size, suggesting

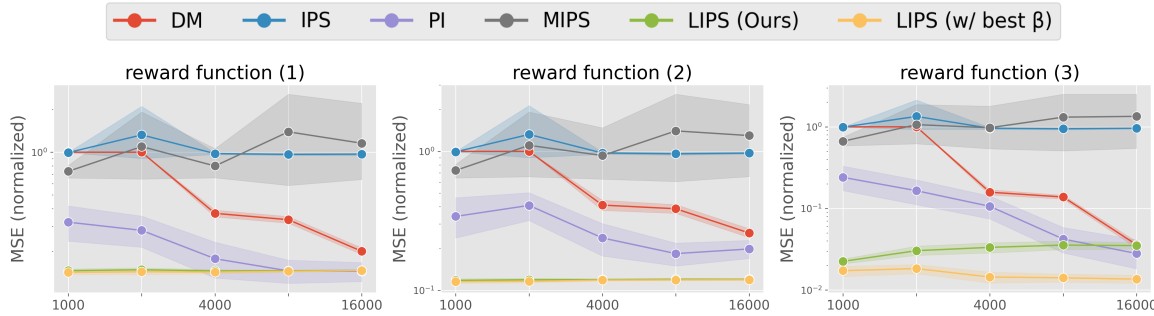

**Figure 3: Comparison of the estimators' MSE (normalized by the true value $V(\pi)$) with varying data sizes ($n$) and with reward functions (1) - (3) on the Wiki10-31K dataset.**

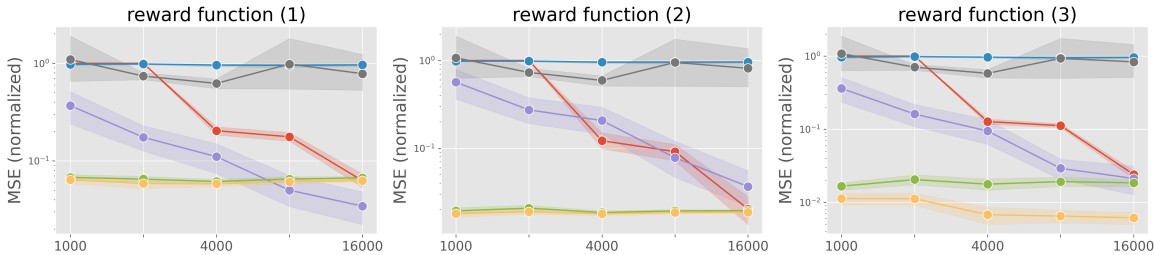

**Figure 4: Comparison of the estimators' MSE (normalized by the true value $V(\pi)$) with varying data sizes ($n$) and with reward functions (1) - (3) on the Eurlex-4K dataset.**

that they need even larger logged dataset to be effective. Finally, LIPS performs much better than other estimators in most cases with a feasible hyper-parameter selection by SLOPE, providing a further argument about its effectiveness for non-linear reward functions.[3]

Beyond basic comparisons against DM, IPS, PI, and MIPS, we finally compare LIPS with a set of hybrid estimators including DR [10], PI-DR [33], and OffCEM [29] (Appendix A defines these estimators in detail with math notations). Figures 5 and 6 show this comparison on Wiki10-31K and Eurlex-4K for varying slate sizes ($L \in \{4, 6, \ldots, 12\}$). Figures 7 and 8 compare the methods on the datasets for varying data sizes ($n \in \{1000, 2000, \ldots, 16000\}$). The results demonstrate that LIPS outperforms these hybrid estimators for a range of experiment values ($L$ and $n$) and non-linear reward functions. This is because DR has a variance issue due to its slate-wise importance weighting while PI-DR suffers from bias due to its linearity assumption and variance due to its slot-wise importance weighting. OffCEM is an extension of MIPS and uses a sufficient abstraction, but it is not optimized towards the MSE, and thus produces much larger variance than LIPS. Note that we also observe the similar trend on an additional dataset as reported in Appendix A. These empirical observations suggest that, for OPE of slate bandits, tuning the definition of importance weights by optimizing slate abstractions with our method is more crucial and effective than adding a reward estimator ($\hat{q}$) as done in the hybrid estimators.

## 6 CONCLUSION AND FUTURE WORK

This paper studied OPE of slate contextual bandits where existing estimators encounter significant challenges in terms of bias and variance due to combinatorial action spaces and restrictive assumptions on the reward function. To overcome these limitations, we proposed LIPS, a novel estimator that is built on slate abstraction to substantially reduce variance. Our analysis demonstrates that LIPS can be unbiased given a sufficient abstraction, and provides a substantial reduction in variance. We also observed that the advantages of LIPS may be maximized when we intentionally use an insufficient abstraction. Based on this analysis, we presented a method for optimizing slate abstraction to minimize the bias and variance of LIPS, which differentiates our work from MIPS [28] and OffCEM [29]. Our experiments illustrate that LIPS enables considerably more accurate OPE for slate actions than existing estimators, including IPS, PI, MIPS, and their DR variants.

Our work also highlights several promising directions for future research. First, LIPS can also be applied to OPE of ranking policies where each slot has observable reward [16, 21]. Furthermore, it would be practically valuable to incorporate control variates into LIPS to further improve its variance, which is considered a non-trivial extension since it would change the way we should optimize slate abstractions. Lastly, it would be intriguing to explore the application of recent advances in diffusion models [2, 45] to optimize slate abstractions more effectively.

---

[3]In addition to Wiki10-31K and Eurlex-4K, we performed the same experiments with varying slate and data sizes on an additional dataset and observed similar trends as reported in Appendix A.

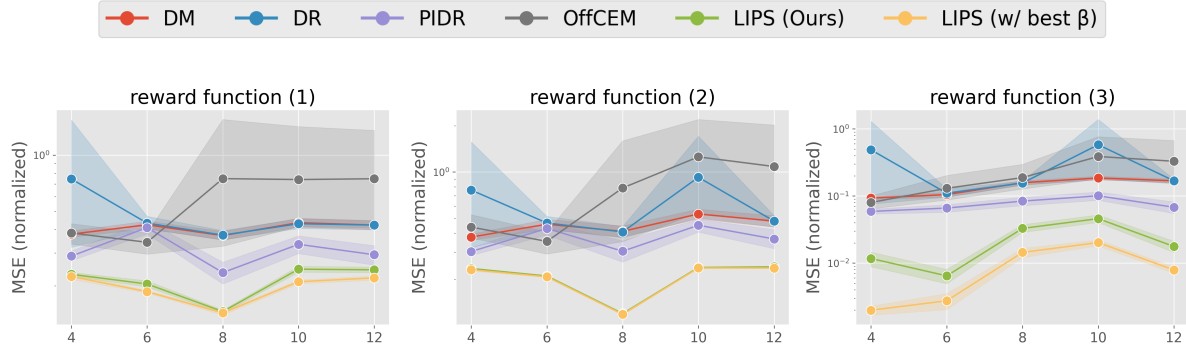

**Figure 5: Comparison between LIPS and DR estimators wrt their MSE (normalized by the true value $V(\pi)$) under varying slate sizes ($L$) and with reward functions (1) - (3) on the Wiki10-31K dataset.**

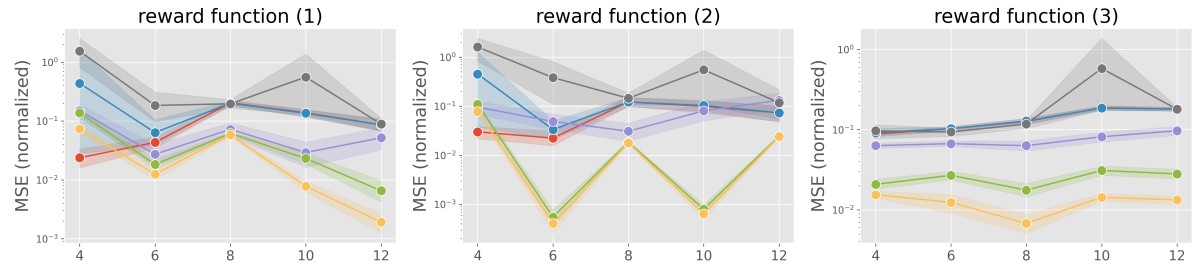

**Figure 6: Comparison between LIPS and DR estimators wrt their MSE (normalized by the true value $V(\pi)$) under varying slate sizes ($L$) and with reward functions (1) - (3) on the Eurlex-4K dataset.**

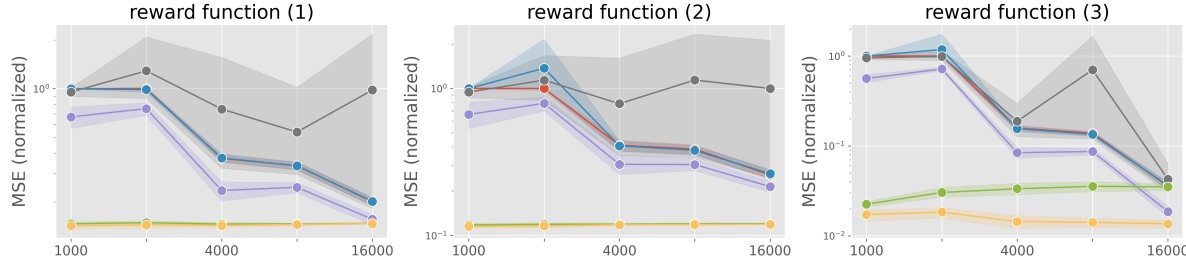

**Figure 7: Comparison between LIPS and DR estimators wrt their MSE (normalized by the true value $V(\pi)$) under varying data sizes ($n$) and with reward functions (1) - (3) on the Wiki10-31K dataset.**

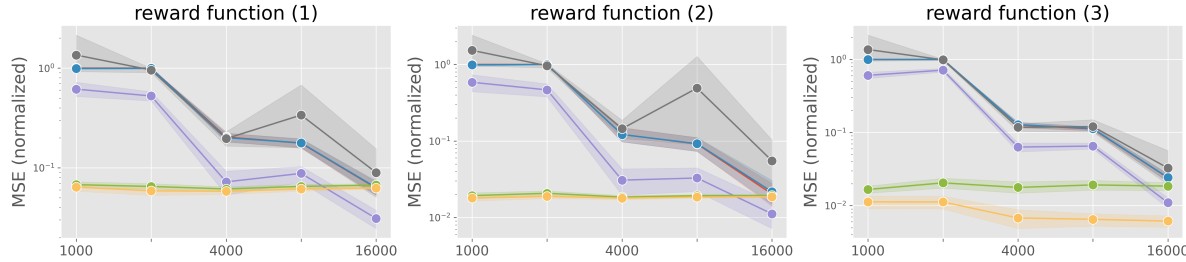

**Figure 8: Comparison between LIPS and DR estimators wrt their MSE (normalized by the true value $V(\pi)$) under varying data sizes ($n$) and with reward functions (1) - (3) on the Eurlex-4K dataset.**

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

---

**Algorithm 1** Optimization procedure for slate abstraction used in LIPS

---

**Input:** logged data $\mathcal{D}$, bias-variance tradeoff hyper-parameter $\beta$, learning rates for slate abstraction model $\tau_\theta$, slate reconstruction model $\tau_\psi$, and reward construction model $\tau_\omega$, maximum gradient steps $T$, batch size $B$

**Output:** optimized slate abstraction distribution $p_\theta(z \mid \boldsymbol{x}, \boldsymbol{s})$

1: Initialize the parameters of the slate abstraction model, slate reconstruction model, and reward construction model $(\theta, \psi, \omega)$
2: **for** $t \in \{1, 2, \cdots, T\}$ **do**
3:   Sample size $B$ of mini-batch data $\mathcal{D}_B^{(t)} \sim \mathcal{D}$
4:   **for** $i \in \{1, 2, \cdots, B\}$ **do**
5:     Retrieve data tuple $(\boldsymbol{x}_i, \boldsymbol{s}_i, r_i) \sim \mathcal{D}_B^{(t)}$
6:     Sample a slate abstraction as $z_i \sim p_\theta(\cdot \mid \boldsymbol{x}_i, \boldsymbol{s}_i)$
7:   **end for**
8:   Compute the slate reconstruction loss as
     $\hat{\mathcal{L}}_{b1}(\boldsymbol{x}, \boldsymbol{s}, \pi_0; \theta, \psi) = \frac{1}{B} \sum_{i \in [B]} \log p_\psi(\boldsymbol{s}_i \mid \boldsymbol{x}_i, z_i)$
9:   Compute the reward construction loss as
     $\hat{\mathcal{L}}_{b2}(\boldsymbol{x}, \boldsymbol{s}, \pi_0; \theta, \omega) = \frac{1}{B} \sum_{i \in [B]} (r_i - \hat{q}_\omega(\boldsymbol{x}_i, z_i))^2$
10:  Compute the KL (regularization) loss as
     $\hat{\mathcal{L}}_{v1}(\boldsymbol{x}, \boldsymbol{s}, \pi_0; \theta) = \frac{1}{B} \sum_{i \in [B]} (\log p_\theta(z_i \mid \boldsymbol{x}_i, \boldsymbol{s}_i) - C)$
     (where $C = -\log(|\mathcal{Z}|)$ is the log of the prior distribution of $p_\psi(z \mid \boldsymbol{x}, \boldsymbol{s}) = |\mathcal{Z}|^{-1}$)
11:  Compute the total loss as
     $\hat{\mathcal{L}}(\boldsymbol{x}, \boldsymbol{s}, \pi_0; \theta, \psi, \omega) = \hat{\mathcal{L}}_{b1}(\boldsymbol{x}, \boldsymbol{s}, \pi_0; \theta, \psi) + \hat{\mathcal{L}}_{b2}(\boldsymbol{x}, \boldsymbol{s}, \pi_0; \theta, \omega) - \beta \hat{\mathcal{L}}_{v1}(\boldsymbol{x}, \boldsymbol{s}, \pi_0; \theta)$
12:  Update the parameters of the slate abstraction and reconstruction models
     $\theta_t \leftarrow \theta_{t-1} + \tau_\theta \nabla_\theta \hat{\mathcal{L}}(\boldsymbol{x}, \boldsymbol{s}, \pi_0; \theta, \psi, \omega)$
     $\psi_t \leftarrow \psi_{t-1} + \tau_\psi \nabla_\psi \hat{\mathcal{L}}(\boldsymbol{x}, \boldsymbol{s}, \pi_0; \theta, \psi, \omega)$
13:  Update the parameter of the reward construction model
     $\omega_t \leftarrow \omega_{t-1} - \tau_\omega \nabla_\omega \hat{\mathcal{L}}_{b2}(\boldsymbol{x}, \boldsymbol{s}, \pi_0; \theta, \omega)$
14: **end for**

---

# A EXPERIMENTAL DETAILS AND ADDITIONAL RESULTS

We describe the detailed experimental settings and additional results omitted in the main text.

## A.1 Baseline estimators

Below, we summarize the definition and properties of the baseline estimators.

**Direct Method (DM) [4]** DM is a model-based approach, which first trains a reward predictor $\hat{q}(\boldsymbol{x}, \boldsymbol{s}) \approx \mathbb{E}[r|\boldsymbol{x}, \boldsymbol{s}]$ and then estimates the policy performance of $\pi$ using $\hat{q}$ as follows.

$$\hat{V}_{\text{DM}}(\pi; \mathcal{D}) := \frac{1}{n} \sum_{i=1}^{n} \sum_{\boldsymbol{s} \in \mathcal{S}} \pi(\boldsymbol{s}|\boldsymbol{x}_i)\hat{q}(\boldsymbol{x}_i, \boldsymbol{s}) \left( = \frac{1}{n} \sum_{i=1}^{n} \mathbb{E}_{\boldsymbol{s} \sim \pi(\boldsymbol{s}|\boldsymbol{x}_i)}[\hat{q}(\boldsymbol{x}_i, \boldsymbol{s})] \right)$$

The accuracy of DM depends on the accuracy of $\hat{q}(\boldsymbol{x}, \boldsymbol{s})$, and the prediction error of $\hat{q}(\boldsymbol{x}, \boldsymbol{s})$ introduces bias and makes DM no longer consistent. Oftentimes, DM has high bias because overfitting and model mis-specification are not easily detectable when training $\hat{q}(\boldsymbol{x}, \boldsymbol{s})$ on logged data.

**Inverse Propensity Scoring (IPS) [32]** As described in the main text, IPS applies the importance sampling technique to reweigh the observed rewards and correct the distribution shift between $\pi$ and $\pi_0$ as follows.

$$\hat{V}_{\text{IPS}}(\pi; \mathcal{D}) := \frac{1}{n} \sum_{i=1}^{n} \frac{\pi(\boldsymbol{s}_i|\boldsymbol{x}_i)}{\pi_0(\boldsymbol{s}_i|\boldsymbol{x}_i)} r_i = \frac{1}{n} \sum_{i=1}^{n} \left( \prod_{l=1}^{L} \frac{\pi(a_{i,l}|\boldsymbol{x}_i)}{\pi_0(a_{i,l}|\boldsymbol{x}_i)} \right) r_i$$

IPS is unbiased under some identification assumption. However, it has exponential variance due to combinatorially large slate space [36].

**PseudoInverse (PI) [36]** PI assumes that the expected reward is linearly attributed to each slot as $q(\boldsymbol{x}, \boldsymbol{s}) = \sum_{l=1}^{L} \phi_l(\boldsymbol{x}, a_l)$, where $\{\phi_l\}_{l=1}^{L}$ is some (latent) intrinsic reward function. Based on this assumption, PI corrects the distribution shift of $\pi$ and $\pi_0$ by only applying slot-level

importance sampling, as shown in the main text.

$$\hat{V}_{\text{PI}}(\pi; \mathcal{D}) := \frac{1}{n} \sum_{i=1}^{n} \left( \sum_{l=1}^{L} \frac{\pi(a_{i,l}|\boldsymbol{x}_i)}{\pi_0(a_{i,l}|\boldsymbol{x}_i)} - L + 1 \right) r_i$$

PI is unbiased when the linearity assumption holds, while it introduces non-negligible bias when the assumption does not hold. In addition, while the variance of PI is much smaller than IPS, PI may still suffer from a high variance when there are many unique sub-actions and $\sum_{l=1}^{L} |\mathcal{A}_l|$ becomes large [28].

**Marginal IPS (MIPS) [28]**   MIPS is originally defined in a contextual bandit setting where some action embeddings (i.e., the genre of a video or sentiment of a movie) are observed. Specifically, MIPS considers the following data generation process:

$$(\boldsymbol{x}, a, \boldsymbol{e}, r) \sim p(\boldsymbol{x})\pi(a|\boldsymbol{x})p(\boldsymbol{e}|a)p(r|\boldsymbol{x}, a, \boldsymbol{e}) \tag{10}$$

where $\boldsymbol{x} \in \mathcal{X}$ is a context, $a \in \mathcal{A}$ is an action, $\boldsymbol{e} \in \mathcal{E}$ is an embedding (vector), and $r \in \mathbb{R}$ is a reward. We should note here that the embeddings in (10) are strictly different from abstractions defined in our formulation in the main text. First, embeddings are observed in the logged data, while our abstraction is unobservable. Moreover, whereas embeddings are sampled from a pre-defined embedding distribution $p(\boldsymbol{e}|\boldsymbol{a})$, even a "true" abstraction distribution does not exist and $p(z|\boldsymbol{x}, \boldsymbol{s})$ should rather be optimized in our problem setting. Then, given the data generation process described in (10), MIPS applies the importance sampling on the embedding space $\mathcal{E}$ as follows.

$$\hat{V}_{\text{MIPS}}^{(\text{original})} := \frac{1}{n} \sum_{i=1}^{n} \frac{\pi(\boldsymbol{e}_i|\boldsymbol{x}_i)}{\pi_0(\boldsymbol{e}_i|\boldsymbol{x}_i)} r_i = \frac{1}{n} \sum_{i=1}^{n} \frac{\sum_{a \in \mathcal{A}} \pi(a|\boldsymbol{x}_i)p(\boldsymbol{e}_i|a)}{\sum_{a \in \mathcal{A}} \pi_0(a|\boldsymbol{x}_i)p(\boldsymbol{e}_i|a)} r_i$$

where $\pi(\boldsymbol{e}|\boldsymbol{x}) = \sum_{a \in \mathcal{A}} \pi(a|\boldsymbol{x})p(\boldsymbol{e}|a)$ is the marginal probability of a policy $\pi$ chooses the actions associated with the embedding $\boldsymbol{e}$. MIPS is unbiased when the following *no direct effect* assumption holds.

ASSUMPTION A.1.   *(Assumption 3.2 of Saito and Joachims [28]) Action a has no direct effect on reward, i.e., $a \perp r \mid x, \boldsymbol{e}$.*

Assumption A.1 means that the reward distribution is expressed as $p(r|x, a, e) = p(r|x, e)$. While the (slate) embeddings are unavailable in our setting, a *sufficient slate abstraction* defined in Definition 3.1 satisfies the no direct effect assumption (Assumption A.1) if we regard $\phi(\boldsymbol{s})$ as a pseudo-embedding. Therefore, in our experiment, we regard the following estimator, which satisfies the no direct effect assumption, as MIPS, given that $q(\boldsymbol{x}, \boldsymbol{s})$ is only affected by $\tilde{\boldsymbol{s}} = (a_1, a_2, \cdots, a_{\lfloor L/2 \rfloor})$.

$$\hat{V}_{\text{MIPS}}(\pi; \mathcal{D}) := \frac{1}{n} \sum_{i=1}^{n} \frac{\pi(\tilde{\boldsymbol{s}}_i|\boldsymbol{x}_i)}{\pi_0(\tilde{\boldsymbol{s}}_i|\boldsymbol{x}_i)} r_i \left( = \frac{1}{n} \sum_{i=1}^{n} \left( \prod_{l=1}^{\lfloor L/2 \rfloor} \frac{\pi(a_{i,l}|\boldsymbol{x}_i)}{\pi_0(a_{i,l}|\boldsymbol{x}_i)} \right) r_i \right)$$

MIPS is unbiased and reduces the variance compared to IPS. However, its variance remains high because the combinatorial action space can still be large even when we consider the sufficient slate space of $\tilde{\boldsymbol{s}}$ rather than the original slate space of $\boldsymbol{s}$.

**Doubly Robust (DR) [9]**   DR is a hybrid estimator that combines both model-based and importance sampling-based approaches. Specifically, DR uses $\hat{q}(\boldsymbol{x}, \boldsymbol{s})$ as a control variate and applies importance sampling only on the residual to reduce the variance as follows.

$$\hat{V}_{\text{DR}}(\pi; \mathcal{D}) := \frac{1}{n} \sum_{i=1}^{n} \left\{ \frac{\pi(\boldsymbol{s}_i|\boldsymbol{x}_i)}{\pi_0(\boldsymbol{s}_i|\boldsymbol{x}_i)} (r_i - \hat{q}(\boldsymbol{x}_i, \boldsymbol{s}_i)) + \sum_{\boldsymbol{s} \in \mathcal{S}} \pi(\boldsymbol{s}|\boldsymbol{x}_i)\hat{q}(\boldsymbol{x}_i, \boldsymbol{s}) \right\}$$

$$= \frac{1}{n} \sum_{i=1}^{n} \left( \prod_{l=1}^{L} \frac{\pi(a_{i,l}|\boldsymbol{x}_i)}{\pi_0(a_{i,l}|\boldsymbol{x}_i)} \right) (r_i - \hat{q}(\boldsymbol{x}_i, \boldsymbol{s}_i)) + \hat{V}_{\text{DM}}(\pi; \mathcal{D})$$

DR is unbiased and reduces the variance of IPS when $|\hat{q}(\boldsymbol{x}, \boldsymbol{s}) - q(\boldsymbol{x}, \boldsymbol{s})| \leq q(\boldsymbol{x}, \boldsymbol{s})$ holds for any $(\boldsymbol{x}, \boldsymbol{s}) \in \mathcal{X} \times \mathcal{S}$. However, DR can still suffer from high variance when the importance weight is large or $\hat{q}(\boldsymbol{x}, \boldsymbol{s})$ is inaccurate [28].

**PI-DR [33, 41]**   PI-DR is a DR-variant of PI, which estimates the policy performance as follows.

$$\hat{V}_{\text{PI-DR}}(\pi; \mathcal{D}) := \frac{1}{n} \sum_{i=1}^{n} \left\{ \left( \sum_{l=1}^{L} \frac{\pi(a_{i,l}|\boldsymbol{x}_i)}{\pi_0(a_{i,l}|\boldsymbol{x}_i)} - L + 1 \right) (r_i - \hat{q}(\boldsymbol{x}_i, \boldsymbol{s}_i)) + \sum_{\boldsymbol{s} \in \mathcal{S}} \pi(\boldsymbol{s}|\boldsymbol{x}_i)\hat{q}(\boldsymbol{x}_i, \boldsymbol{s}) \right\}$$

$$= \frac{1}{n} \sum_{i=1}^{n} \left( \sum_{l=1}^{L} \frac{\pi(a_{i,l}|\boldsymbol{x}_i)}{\pi_0(a_{i,l}|\boldsymbol{x}_i)} - L + 1 \right) (r_i - \hat{q}(\boldsymbol{x}_i, \boldsymbol{s}_i)) + \hat{V}_{\text{DM}}(\pi; \mathcal{D})$$

PI-DR is unbiased when the linearity assumption holds. However, its bias becomes high when the assumption does not hold. PI-DR also reduces the variance of PI under a reasonable assumption about reward prediction ($|\hat{q}(\boldsymbol{x}, \boldsymbol{s}) - q(\boldsymbol{x}, \boldsymbol{s})| \leq q(\boldsymbol{x}, \boldsymbol{s})$), while the variance problem can remain when $\sum_{l=1}^{L} |\mathcal{A}_l|$ is large or $\hat{q}(\boldsymbol{x}, \boldsymbol{s})$ is inaccurate.

**OffCEM [29]**   OffCEM is another hybrid estimator that combines model-based and importance sampling-based approaches building on the MIPS estimator as follows.

$$\hat{V}_{\text{OffCEM}}(\pi; \mathcal{D}) := \frac{1}{n} \sum_{i=1}^{n} \left\{ \frac{\pi(\tilde{s}_i|x_i)}{\pi_0(\tilde{s}_i|x_i)} (r_i - \hat{q}(x_i, s_i)) + \sum_{s \in \mathcal{S}} \pi(s|x_i)\hat{q}(x_i, s) \right\}$$

$$= \frac{1}{n} \sum_{i=1}^{n} \left( \prod_{l=1}^{\lfloor L/2 \rfloor} \frac{\pi(a_{i,l}|x_i)}{\pi_0(a_{i,l}|x_i)} \right) (r_i - \hat{q}(x_i, s_i)) + \hat{V}_{\text{DM}}(\pi; \mathcal{D})$$

OffCEM is unbiased either when *no direct effect* assumption holds about $\tilde{s}$ or when the reward predictor accurately captures the pair-wise value difference between two slates within the same slate cluster (i.e., $\hat{q}(x, s_1) - \hat{q}(x, s_2) = q(x, s_1) - q(x, s_2), \forall s_1, s_2, \tilde{s}_1 = \tilde{s}_2$). OffCEM also reduces the variance of DR, however, the degree of variance reduction remains small when the slate space of $\tilde{s}$ remains large.

***Cross-fitting technique***  DR-type estimators achieve minimum possible variance among the estimator class building on the same importance weight as used in DR-type estimators, when $\hat{q}(x_i, s_i)$ is consistent and independent with $r_i$ (referred to as *semiparametric efficiency* [24]). Therefore, to avoid potential bias caused by overfitting and derive the consistent estimation of $\hat{q}(x, s)$, we use the following cross-fitting procedure to train $\hat{q}$ used in DM and DR-type estimators.

(1) Given size $n$ of logged data $\mathcal{D}$, randomly split the data into $K$-fold partition $\{\mathcal{D}_\kappa\}_{\kappa=1}^{K}$, each of which contains $n_\kappa = n/K$ samples. Let $\mathcal{D}_\kappa^c := \mathcal{D} \setminus \mathcal{D}_\kappa$ be the subset of $\kappa$-th partition.
(2) For each $\kappa = 1, 2, \cdots K$, train a reward predictors $\hat{q}_\kappa$ on the subset of the $\kappa$-th partition $\mathcal{D}_\kappa^c$.
(3) Estimate the policy performance by $\hat{V}(\pi; \mathcal{D}) = K^{-1} \sum_{\kappa=1}^{K} \hat{V}(\pi; \mathcal{D}_\kappa, \hat{q}_\kappa)$.

## A.2   Hyperparameter tuning via the SLOPE algorithm [34, 38]

To tune the bias-variance tradeoff hyperparameter $\beta$ of our optimization procedure based only on the logged bandit data $\mathcal{D}$, we use the SLOPE algorithm [34, 38]. SLOPE is able to select a suitable hyperparameter $\lambda$ from a candidate set $\Lambda := \{\lambda_m\}_{m=1}^{M}$ as long as an estimator satisfies the following *monotonicity* condition [38].

(1) $\text{Bias}(\hat{V}(\cdot; \lambda_m)) \leq \text{Bias}(\hat{V}(\cdot; \lambda_{m+1})), \forall m \in [M-1]$
(2) $\text{CNF}(\hat{V}(\cdot; \lambda_m)) \geq \text{CNF}(\hat{V}(\cdot; \lambda_{m+1})), \forall m \in [M-1]$

where $\text{CNF}(\cdot)$ is a high probability bound on the deviation of $\hat{V}$ such as the Hoeffding and Bernstein bounds [37]. Note that, LIPS and its hyper-parameter $\beta$ satisfy this monotonicity condition, as we know that a larger value of $\beta$ reduces the variance more, while a smaller value reduces the bias more. Specifically, SLOPE selects the hyperparameter as follows.

$$\hat{m} := \max\{m \in [M] : |\hat{V}(\cdot; \lambda_m) - \hat{V}(\cdot; \lambda_{m'})|$$
$$\leq \text{CNF}(\hat{V}(\cdot; \lambda_m)) + (\sqrt{6} - 1)\text{CNF}(\hat{V}(\cdot; \lambda_{m'})), \forall m' < m\}.$$

When the monotonicity condition holds, SLOPE guarantees that the deviation of $\hat{V}$ is upper bounded as below with probability $1 - \delta$:

$$|\hat{V}(\cdot; \lambda_{\hat{m}}) - \hat{V}(\cdot; \lambda_{m^*})| \leq (\sqrt{6} + 3) \min_{m \in [M]} (\text{Bias}(\hat{V}(\cdot; \lambda_m)) + \text{CNF}(\hat{V}(\cdot; \lambda_m))),$$

where $\hat{m}$ is the selected hyperparameter and $m^*$ is the best hyperparameter among the candidate set. Even when the condition does not hold true, SLOPE guarantees the following looser bound.

$$|\hat{V}(\cdot; \lambda_{\hat{m}}) - \hat{V}(\cdot; \lambda_{m^*})| \leq (\sqrt{6} + 3) \min_{m \in [M]} (\max_{j \leq m} \text{Bias}(\hat{V}(\cdot; \lambda_j)) + \max_{k \leq m} \text{CNF}(\hat{V}(\cdot; \lambda_k))).$$

We refer the reader to [38] for the detailed theoretical analysis of SLOPE.

## A.3   Models and parameters

We use a neural network with a 100-dimensional hidden layer to train $\hat{q}$ for DM. The reward predictor uses Adam [14] to minimize the MSE loss with a learning rate of 1e-2. We also use 80% of the logged dataset for training, while the other 20% for testing. We train the model on the train set for 500 epochs with 10 gradient steps for each. We apply early stopping for $\hat{q}$, when the test loss increases for 5 consequent epochs. Note that since taking the exact expectation over the slate space ($\mathcal{S}$) requires huge computational costs, we approximate the expectation over $\pi(s|x)$ by sampling 1,000 slates following $\pi(s|x)$ for each context $x$.

   We optimize the slate abstraction for LIPS using neural networks with a 100-dimensional hidden layer to parameterize the reward construction, slate abstraction, and slate reconstruction models. All these models are optimized using Adam [14] with a learning rate of 1e-5 for both Wiki10-31K and Eurlex-4K datasets. Note that we re-scale the reward loss in all experiments by multiplying it by 100, with the aim to make the scale of the reward loss similar to that of the slate reconstruction loss. Initially, we train the models for 1000 epochs when $\beta = 0.01$, then tune the models for an additional 500 epochs for $\beta$ values of 0.1, 1.0, and 10.0. The experiments are conducted on an M1

Table 2: Statistics of the Extreme Classification datasets used in the experiments.

| dataset | # of documents | features of documents | # of labels | avg. labels per document |
|---------|----------------|----------------------|-------------|--------------------------|
| Wiki10-31K | 14,146 (6,616) | raw texts | 30,938 | 18.64 |
| Eurlex-4K | 15,539 (3,809) | raw texts | 3,993 | 5.31 |
| Delicious | 12,920 (3,185) | 500 dim. of BoW | 983 | 19.03 |

*Note*: The column "# of documents" describes "# of train samples (# of test samples)" of the datasets. We use the training dataset for performing slate OPE, while the test set is used for training the base classifier $\tilde{q}$ to form a logging policy. The raw texts of the Wiki10-31K and Eurlex-4K datasets are converted to 20-dimensional feature vectors via SentenceTransformer [25] and PCA [1]. Finally, for Wiki10-31K and Eurlex-4K, we extract the top 1,000 dense labels after removing the labels that are relevant to more than 1,000 documents.

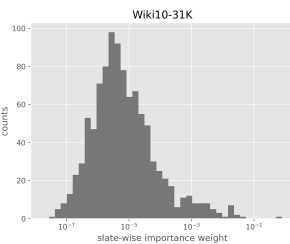 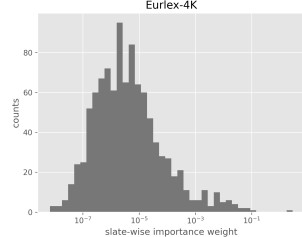 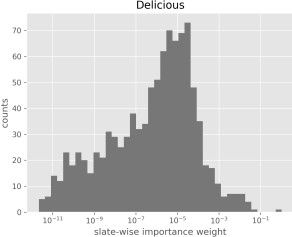

Figure 9: Empirical distributions of the slate-wise importance weight $w(x, s)$ for each dataset used in the experiment; (Left) Wiki10-31K (Center) Eurlex-4K (Right) Delicious.

MacBook Pro, and the optimization process for LIPS takes approximately 3 minutes with a single value of $\beta$ when $L = 4$ and $n = 4000$. When $L = 12$, the computational time increases to about 4.5 minutes, indicating that the runtime increases sub-linearly.

## A.4 Additional experiments on the Delicious dataset

In addition to the experiments described in the main text, we also compare estimators using the Delicious dataset provided in the Extreme Classification repository [5]. The Delicious dataset consists of documents associated with approximately 1K labels, and the detailed statistics of the dataset are described in Table 2. We basically follow the same "supervised-to-bandit" procedure as described in the main text. The only difference is that we use 500-dimensional Bag of Words (BoW) features as contexts because the Delicious dataset does not contain raw texts.

Figures 10-13 show the comparison of LIPS and IPS-based estimators (Figures 10 and 12) and DR-based estimators (Figures 11 and 13) with varying slate sizes $L$ (Figures 10 and 11) and data sizes $n$ (Figures 12 and 13) on the Delicious dataset. The result demonstrates similar trends to those observed in the main text; LIPS (Ours, w/ data-driven choice of $\beta$) often performs the best among the compared estimators, while IPS, PI, and MIPS often produce high estimation error and DM is accurate only when the reward prediction is accurate by chance.[4] These results suggest that LIPS is able to flexibly control the bias-variance tradeoff in a way to minimize MSE on various datasets.

## A.5 Additional ablation experiments with varying values of $\beta$

We also conduct an ablation study of LIPS with varying values of $\beta$ ($\in \{0.01, 0.05, 0.1, 0.5, 1.0, 5.0, 10.0\}$) on Wiki10-31K and Eurlex-4K datasets where we use the default setting of $L = 8$ and $n = 4,000$. Figures 14 and 15 show the MSE, squared bias, and variance of LIPS with each value of $\beta$, the data-driven choice (i.e., LIPS (Ours)), and the best one, respectively. The result demonstrates that there is some bias-variance tradeoff in $\beta \in \{1.0, 5.0, 10.0\}$ while there is some instability in $\beta \leq 1.0$. Specifically, we observe in both the Wiki10-31K and Eurlex-4K datasets that a small value of $\beta$ (i.e., $\beta = 1.0$) reduces bias while having some variance, while a large value of $\beta$ (e.g., $\beta = 5.0, 10.0$) reduces the variance more while introducing some additional bias. These observations align with the theoretical analysis discussed in Section 3.1 and 3.2. The result also shows that there is some room for improvement in the data-driven selection of $\beta$, however, our slate abstraction optimization enables accurate estimation across various values of $\beta$ and demonstrates robust performance even when we choose the value of $\beta$ using a data-driven manner, as shown in the main text. We attribute this observation to the ability of our slate abstraction model to identify a well-optimized slate abstraction that reduces both bias and variance at the same time.

---

[4]For the experiment with varying data sizes, we observe that $\hat{q}$ somehow gradually increases as the data size increases. As a result, DM is accurate for $n = 4000$ and $n = 8000$, but is not accurate for other configurations. It should be worth noting that verifying if $\hat{q}$ is accurate is itself quite challenging in OPE, as the dataset contains reward signals only for the action chosen by the logging policy (i.e., partial rewards).

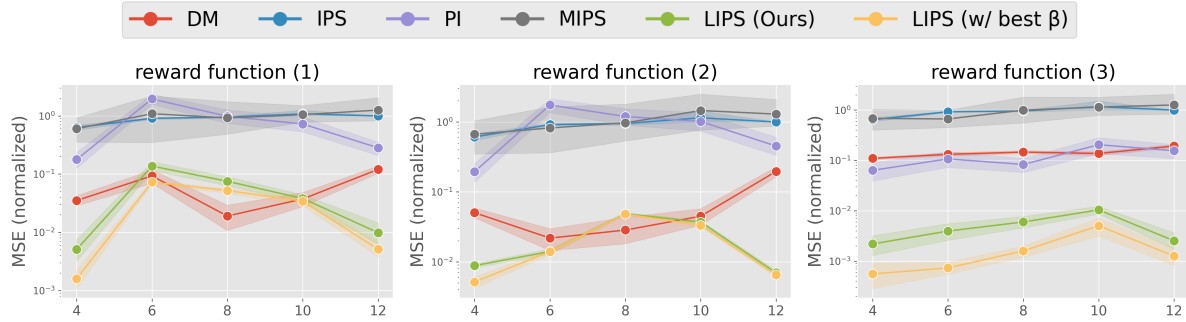

**Figure 10: Comparison of the estimators' MSE (normalized by the true value $V(\pi)$) with varying slate sizes ($L$) and with reward functions (1) - (3) on the Delicious dataset.**

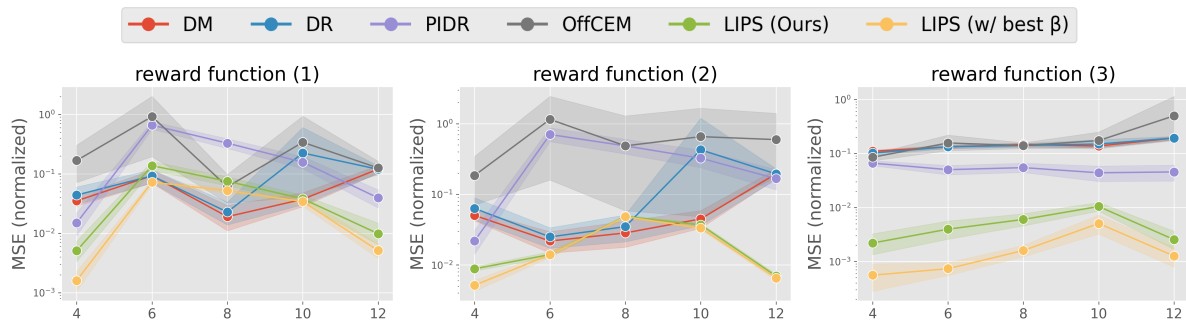

**Figure 11: Comparing the LIPS' MSE (normalized by the true value $V(\pi)$) with those of DR estimators under varying slate sizes ($L$) and with reward functions (1) - (3) on the Delicious dataset.**

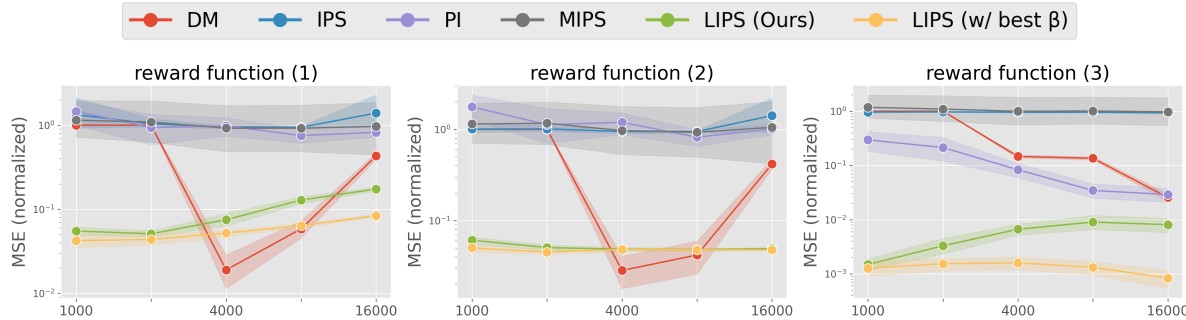

**Figure 12: Comparison of the estimators' MSE (normalized by the true value $V(\pi)$) with varying data sizes ($n$) and with reward functions (1) - (3) on the Delicious dataset.**

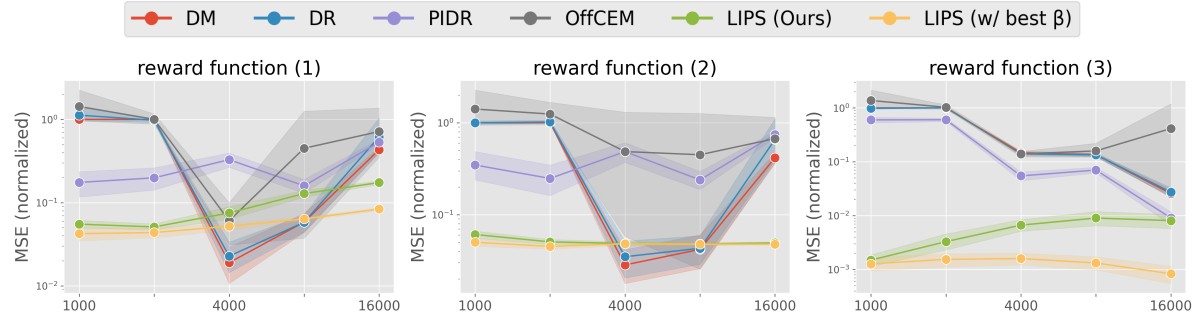

**Figure 13: Comparing the LIPS' MSE (normalized by the true value $V(\pi)$) with those of DR estimators under varying data sizes ($n$) and with reward functions (1) - (3) on the Delicious dataset.**

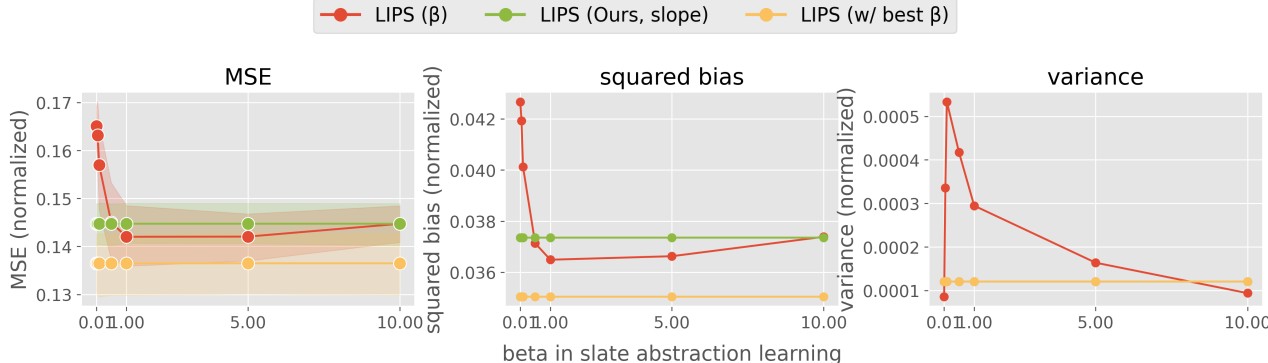

**Figure 14: Comparison of the LIPS' MSE, squared bias, and variance (normalized by the true value $V(\pi)$) with varying values of $\beta$ with reward function (1) on the Wiki10-31K dataset**

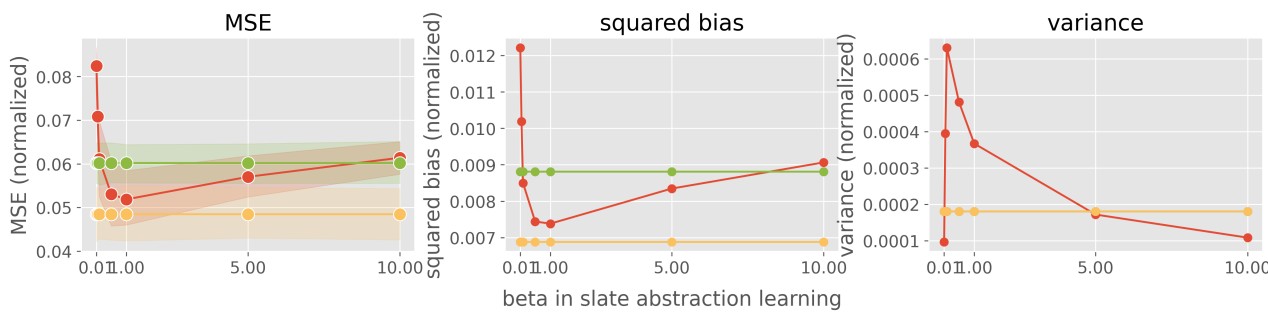

**Figure 15: Comparison of the LIPS' MSE, squared bias, and variance (normalized by the true value $V(\pi)$) with varying values of $\beta$ with reward function (1) on the Eurlex-4K dataset.**

# B DETAILED BIAS-VARIANCE ANALYSIS OF LIPS AND OMITTED PROOFS

Here, we provide detailed discussion of bias-variance properties of LIPS and the proofs of Theorems.

## B.1 Preliminaries

As a warm-up, we first derive an alternative expression of the latent importance weight as follows:

$$
\begin{aligned}
w_\theta(x, z) &= \frac{p_\theta(z \mid x, \pi)}{p_\theta(z \mid x, \pi_0)} \\
&= \frac{\sum_{s \in \mathcal{S}} p_\theta(z \mid x, s) \pi(s \mid x)}{p_\theta(z \mid x, \pi_0)} \\
&= \frac{\sum_{s \in \mathcal{S}} \frac{p_\theta(z \mid x, \pi_0) p_\theta(s \mid x, z, \pi_0)}{\pi_0(s \mid x)} \pi(s \mid x)}{p_\theta(z \mid x, \pi_0)} \\
&= \sum_{s \in \mathcal{S}} p_\theta(s \mid x, z, \pi_0) \frac{\pi(s \mid x)}{\pi_0(s \mid x)} \\
&= \mathbb{E}_{p_\theta(s \mid x, z, \pi_0)} [w(x, s)]
\end{aligned}
\tag{11}
$$

This means that $w_\theta(x, z)$ can be written as the conditional expectation of $w(x, s)$. Using this expression, we show how our optimization procedure for slate abstraction can balance the bias-variance tradeoff in the following subsection.

## B.2 Bias-variance control of LIPS with the hyper-parameter $\beta$

In this section, we demonstrate that LIPS effectively interpolates between IPS and the naive average estimator (NAE). NAE is defined as the naive empirical average of the observed rewards: $\hat{V}_{\text{NAE}}(\pi; \mathcal{D}) := \frac{1}{n} \sum_{i=1}^{n} r_i$. As NAE does not accommodate the distribution shift between $\pi_0$ and $\pi$, it often leads to a high bias. Contrarily, NAE achieves a much lower variance than IPS because it does not depend on importance weighting. As a result, IPS and NAE exhibit opposite characteristics in terms of the bias-variance tradeoff.

Next, given a sufficiently large abstraction space $\mathcal{Z}$, LIPS becomes identical to IPS and NAE in the special cases as follows.

- When $\beta = 0$, LIPS becomes equal to IPS.
- As $\beta \to +\infty$, LIPS gets close to NAE.

When $\beta = 0$, the abstraction optimization procedure ignores the regularization term (the third term). Therefore, LIPS optimizes a slate abstraction so that it can distinguish the slates accurately, leading to a one-to-one mapping between the slate and abstraction (i.e., both $p_\theta(z|x, s, \pi_0)$ and $p_\psi(s|x, z, \pi_0)$ becomes extremely close to either 0 or 1). This will make our latent importance weight equivalent to the slate-wise importance weight of IPS, and LIPS will be identical to IPS under such a condition. On the other hand, as $\beta \to +\infty$, the KL loss becomes more dominant. In such a situation, $p_\theta(z \mid x, s, \pi_0)$ becomes close to a (pre-defined) prior distribution $p_\psi(z \mid x, \pi_0)$, which leads to the following relationship about the slate abstraction distribution:

$$p_\theta(s \mid x, z, \pi_0) = \pi_0(s|x) \frac{p_\theta(z \mid x, s, \pi_0)}{p_\theta(z \mid x, \pi_0)} \approx \pi_0(s|x) \frac{p_\psi^{(prior)}(z \mid x, \pi_0)}{p_\psi^{(prior)}(z \mid x, \pi_0)} = \pi_0(s|x)$$

By combining this with the alternative expression of the latent importance weight derived in Appendix B.1, we have the following.

$$w_\theta(x, z) = \mathbb{E}_{p_\theta(s \mid x, z, \pi_0)}[w(x, s)] \approx \mathbb{E}_{\pi_0(s \mid a)}[w(x, s)] = \sum_{s \in \mathcal{S}} \pi_0(s \mid x) \frac{\pi(s \mid x)}{\pi_0(s \mid x)} = 1$$

This means that LIPS applies no correction, i.e., $w_\theta(x, z) = 1$, when $\beta \to +\infty$.

To summarize, by adjusting $\beta \in [0, +\infty)$, LIPS interpolates between IPS and NAE in a way that minimizes the MSE of LIPS, without introducing any structural assumptions on the reward function.

## B.3 Proof of Theorem 3.2

The following provides the proof of Theorem 3.2.

PROOF. Given sufficient slate abstraction $\phi_\theta(\cdot)$ s.t., $\forall s \in \mathcal{S}, q(x, s) = q(x, \phi_\theta(s))$, we have the following.

$$
\begin{aligned}
\mathbb{E}_{\mathcal{D}}[\hat{V}_{\mathrm{LIPS}}(\pi; \mathcal{D})] &= \mathbb{E}_{\mathcal{D}}[w_\theta(x, \phi_\theta(s)) \, r] \\
&= \mathbb{E}_{p(x)\pi_0(s \,|\, x)}[w_\theta(x, \phi_\theta(s)) \, q(x, s)] \\
&= \mathbb{E}_{p(x)\pi_0(s \,|\, x)}[w_\theta(x, \phi_\theta(s)) \, q(x, \phi_\theta(s))] \\
&= \mathbb{E}_{p(x)}\Big[\sum_{s \in \mathcal{S}} \pi_0(s|x) \frac{\pi(\phi_\theta(s) \,|\, x)}{\pi_0(\phi_\theta(s) \,|\, x)} q(x, \phi_\theta(s))\Big] \\
&= \mathbb{E}_{p(x)}\Big[\sum_{s \in \mathcal{S}} \pi_0(\phi_\theta(s) \,|\, x) \frac{\pi_0(s|x)}{\pi_0(\phi_\theta(s) \,|\, x)} \frac{\pi(\phi_\theta(s) \,|\, x)}{\pi_0(\phi_\theta(s) \,|\, x)} q(x, \phi_\theta(s))\Big] \\
&= \mathbb{E}_{p(x)}\Big[\sum_{s \in \mathcal{S}} \pi(\phi_\theta(s) \,|\, x)\pi_0(s \,|\, x, \phi_\theta(s)) \, q(x, \phi_\theta(s))\Big] \\
&= \mathbb{E}_{p(x)}\Big[\sum_{z \in \mathcal{Z}} \pi(z \,|\, x) \sum_{s \in \{s' \in \mathcal{S} | \phi_\theta(s')=z\}} \pi_0(s \,|\, x, z) \, q(x, z)\Big] \\
&= \mathbb{E}_{p(x)}\Big[\sum_{z \in \mathcal{Z}} \pi(z \,|\, x) q(x, z)\Big] \\
&= \mathbb{E}_{p(x)}\Big[\sum_{z \in \mathcal{Z}} \pi(z \,|\, x) \sum_{s \in \{s' \in \mathcal{S} | \phi_\theta(s')=z\}} \pi(s \,|\, x, z) \, q(x, z)\Big] \\
&= \mathbb{E}_{p(x)}\Big[\sum_{s \in \mathcal{S}} \pi(\phi_\theta(s) \,|\, x)\pi(s \,|\, x, \phi_\theta(s)) \, q(x, \phi_\theta(s))\Big] \\
&= \mathbb{E}_{p(x)}\Big[\sum_{s \in \mathcal{S}} \pi(s \,|\, x) \, q(x, \phi_\theta(s))\Big] \\
&= \mathbb{E}_{p(x)}\Big[\sum_{s \in \mathcal{S}} \pi(s \,|\, x) \, q(x, s)\Big] \\
&= V(\pi)
\end{aligned}
$$

$\square$

Note that we use $\phi_\theta(s) \, (= z) \in \mathcal{Z}$ and $\mathcal{S} = \{\bigcup_{z \in \mathcal{Z}} \bigcup_{s \in \{s' \in \mathcal{S} | \phi_\theta(s')=z\}} s\}$.

## B.4 Proof of Theorem 3.3

To prove Theorem 3.3, we first import the following lemma from [28].

LEMMA B.1. *(Lemma B.1. of [28]) For real-valued, bounded functions $f : \mathbb{N} \to \mathbb{R}, g : \mathbb{N} \to \mathbb{R}, h : \mathbb{N} \to \mathbb{R}$ where $\sum_{a \in [m]} g(a) = 1$, we have*

$$
\sum_{a \in [m]} f(a)g(a)\Big(h(a) - \sum_{b \in [m]} g(b)h(b)\Big) = \sum_{a < b \leq m} g(a)g(b)(h(a) - h(b))(f(a) - f(b))
$$

Then, we provide the proof in the following.

PROOF. We show this in the case of discrete slate abstraction. Similar proofs hold for continuous slate abstraction by replacing $\sum_{z \in \mathcal{Z}}$ to $\int_{z \in \mathcal{Z}} dz$.

$$\text{Bias}(\hat{V}_{\text{LIPS}}(\pi; \mathcal{D}))$$

$$= \mathbb{E}_{\mathcal{D}}[w_\theta(x, z) r] - V(\pi)$$

$$= \mathbb{E}_{p(x)\pi_0(s|x)p_\theta(z|x,s)}[w_\theta(x, z)q(x, s)] - \mathbb{E}_{p(x)\pi(s|x)p_\theta(z|x,s)}[q(x, s)]$$

$$= \mathbb{E}_{p(x)}\Big[\sum_{s \in \mathcal{S}} \pi_0(s|x) \sum_{z \in \mathcal{Z}} \frac{p_\theta(z|x, \pi_0)p_\theta(s|x, z, \pi_0)}{\pi_0(s|x)} w_\theta(x, z)q(x, s)\Big]$$

$$\quad - \mathbb{E}_{p(x)}\Big[\sum_{s \in \mathcal{S}} \pi(s|x) \sum_{z \in \mathcal{Z}} \frac{p_\theta(z|x, \pi_0)p_\theta(s|x, z, \pi_0)}{\pi_0(s|x)} w_\theta(x, z)q(x, s)\Big]$$

$$= \mathbb{E}_{p(x)}\Big[\sum_{z \in \mathcal{Z}} p_\theta(z|x, \pi_0)w_\theta(x, z) \sum_{s \in \mathcal{S}} p_\theta(s|x, z, \pi_0)q(x, s)\Big]$$

$$\quad - \mathbb{E}_{p(x)}\Big[\sum_{z \in \mathcal{Z}} p_\theta(z|x, \pi_0) \sum_{s \in \mathcal{S}} p_\theta(s|x, z, \pi_0)w(x, s)q(x, s)\Big]$$

$$= \mathbb{E}_{p(x)}\Big[\sum_{z \in \mathcal{Z}} p_\theta(z|x, \pi_0)\Big(\sum_{s \in \mathcal{S}} p_\theta(s|x, z, \pi_0)w(x, s)\Big) \sum_{s' \in \mathcal{S}} p_\theta(s'|x, z, \pi_0)q(x, s')\Big]$$

$$\quad - \mathbb{E}_{p(x)}\Big[\sum_{z \in \mathcal{Z}} p_\theta(z|x, \pi_0) \sum_{s \in \mathcal{S}} p_\theta(s|x, z, \pi_0)w(x, s)q(x, s)\Big]$$

$$= \mathbb{E}_{p(x)p_\theta(z|x,\pi_0)}\Big[\sum_{s \in \mathcal{S}} p_\theta(s|x, z, \pi_0)w(x, s)\Big(\sum_{s' \in \mathcal{S}} p_\theta(s'|x, z, \pi_0)q(x, s') - q(x, s)\Big)\Big]$$

$$= \mathbb{E}_{p(x)p_\theta(z|x,\pi_0)}\Big[\sum_{j<k \leq |\mathcal{S}|} p_\theta(s_j|x, z, \pi_0)p_\theta(s_k|x, z, \pi_0)$$

$$\times (q(x, s_j) - q(x, s_k)) \times (w(x, s_k) - w(x, s_j))\Big]$$

where the last line uses Lemma B.1. □

## B.5 Proof of Theorem 3.4

PROOF. Here, we use the unbiasedness of LIPS under a sufficient slate abstraction (i.e., Theorem 3.2).

$$n\Big(\mathbb{V}_{\mathcal{D}}[\hat{V}_{\text{IPS}}(\pi; \mathcal{D})] - \mathbb{V}_{\mathcal{D}}[\hat{V}_{\text{LIPS}}(\pi; \mathcal{D})]\Big)$$

$$= \mathbb{E}_{\mathcal{D}}[(w(x, s) r - V(\pi))^2] + \mathbb{E}_{\mathcal{D}}[(w_\theta(x, \phi_\theta(s)) r - V(\pi))^2]$$

$$= \mathbb{E}_{p(x)\pi_0(s|x)p(r|x,s)}[(w(x, s) - w(s, \phi_\theta(s)))^2 r^2]$$

$$= \mathbb{E}_{p(x)\pi_0(s|x)}[(w(x, s) - w(s, \phi_\theta(s)))^2 \mathbb{E}_{p(r|x,\phi_\theta(s))}[r^2]]$$

$$= \mathbb{E}_{p(x)}\Big[\sum_{s \in \mathcal{S}} \pi_0(s|x)(w(x, s) - w(s, \phi_\theta(s)))^2 \mathbb{E}_{p(r|x,\phi_\theta(s))}[r^2]\Big]$$

$$= \mathbb{E}_{p(x)}\Big[\sum_{s \in \mathcal{S}} \pi_0(\phi_\theta(s)|x) \frac{\pi_0(s|x)}{\pi_0(\phi_\theta(s)|x)} (w(x, s) - w(s, \phi_\theta(s)))^2 \mathbb{E}_{p(r|x,\phi_\theta(s))}[r^2]\Big]$$

$$= \mathbb{E}_{p(x)}\Big[\sum_{z \in \mathcal{Z}} \pi_0(z|x) \sum_{s \in \{s' \in \mathcal{S}|\phi_\theta(s')=z\}} \pi_0(s|x, z)(w(x, s) - w(s, z))^2 \mathbb{E}_{p(r|x,z)}[r^2]\Big]$$

$$= \mathbb{E}_{p(x)}\Big[\sum_{z \in \mathcal{Z}} \pi_0(z|x) \sum_{s \in \{s' \in \mathcal{S}|\phi_\theta(s')=z\}} \pi_0(s|x, z)$$

$$\cdot \Big(w(x, s) - \sum_{s \in \{s' \in \mathcal{S}|\phi_\theta(s')=z\}} \pi_0(s|x, z)w(x, s)\Big)^2 \mathbb{E}_{p(r|x,z)}[r^2]\Big]$$

$$= \mathbb{E}_{p(x)\pi_0(\phi_\theta(s)|x)}[\mathbb{V}_{\pi_0(s|x,\phi_\theta(s))}(w(x, s)) \mathbb{E}_{p(r|x,z)}[r^2]]$$

□

## B.6 MSE gain of LIPS with stochastic slate abstraction

In summary, LIPS has the following MSE gain over IPS with stochastic slate abstraction.

$$
n(\mathrm{MSE}(\hat{V}_{\mathrm{IPS}}(\pi;\mathcal{D})) - \mathrm{MSE}(\hat{V}_{\mathrm{LIPS}}(\pi;\mathcal{D})))
$$

$$
= \mathbb{E}_{p(\boldsymbol{x})p_\theta(z\,|\,\boldsymbol{x},\pi_0)}\big[\mathbb{V}_{p_\theta(s\,|\,\boldsymbol{x},z,\pi_0)}(w(\boldsymbol{x},s)) \cdot \mathbb{E}_{p_\theta(s\,|\,\boldsymbol{x},z,\pi_0)}\big[\mathbb{E}_{p(r\,|\,\boldsymbol{x},s)}[r^2]\big]\big]
$$

$$
+ \mathbb{E}_{p(\boldsymbol{x})p_\theta(z\,|\,\boldsymbol{x},\pi_0)}\big[\mathrm{Cov}_{p_\theta(s\,|\,\boldsymbol{x},z,\pi_0)}\big(w(\boldsymbol{x},s)^2,\, \mathbb{E}_{p(r\,|\,\boldsymbol{x},s)}[r^2]\big)\big]
$$

$$
+ 2V(\pi)\mathrm{Bias}(\hat{V}_{\mathrm{LIPS}}(\pi;\mathcal{D})) + (1-n)\mathrm{Bias}(\hat{V}_{\mathrm{LIPS}}(\pi;\mathcal{D}))^2
$$

Below, we provide the detailed deviation.

$$
n(\mathrm{MSE}(\hat{V}_{\mathrm{IPS}}(\pi;\mathcal{D})) - \mathrm{MSE}(\hat{V}_{\mathrm{LIPS}}(\pi;\mathcal{D})))
$$

$$
= n(\mathbb{V}_{\mathcal{D}}(\hat{V}_{\mathrm{IPS}}(\pi;\mathcal{D})) - \mathbb{V}_{\mathcal{D}}(\hat{V}_{\mathrm{LIPS}}(\pi;\mathcal{D})) - \mathrm{Bias}(\hat{V}_{\mathrm{LIPS}}(\pi;\mathcal{D}))^2)
$$

$$
= \mathbb{V}_{\mathcal{D}}(\hat{V}_{\mathrm{IPS}}(\pi;\mathcal{D})) - \mathbb{V}_{\mathcal{D}}(\hat{V}_{\mathrm{LIPS}}(\pi;\mathcal{D})) - n\mathrm{Bias}(\hat{V}_{\mathrm{LIPS}}(\pi;\mathcal{D}))^2
$$

$$
= \mathbb{E}_{\mathcal{D}}[(\hat{V}_{\mathrm{IPS}}(\pi;\mathcal{D}) - \mathbb{E}_{\mathcal{D}}[\hat{V}_{\mathrm{IPS}}(\pi;\mathcal{D})])^2] - \mathbb{E}_{\mathcal{D}}[(\hat{V}_{\mathrm{LIPS}}(\pi;\mathcal{D}) - \mathbb{E}_{\mathcal{D}}[\hat{V}_{\mathrm{LIPS}}(\pi;\mathcal{D})])^2]
$$

$$
- n\mathrm{Bias}(\hat{V}_{\mathrm{LIPS}}(\pi;\mathcal{D}))^2
$$

$$
= \mathbb{E}_{\mathcal{D}}[(\hat{V}_{\mathrm{IPS}}(\pi;\mathcal{D}))^2] - (\mathbb{E}_{\mathcal{D}}[\hat{V}_{\mathrm{IPS}}(\pi;\mathcal{D})])^2 - (\mathbb{E}_{\mathcal{D}}[(\hat{V}_{\mathrm{LIPS}}(\pi;\mathcal{D}))^2] - (\mathbb{E}_{\mathcal{D}}[\hat{V}_{\mathrm{LIPS}}(\pi;\mathcal{D})])^2)
$$

$$
- n\mathrm{Bias}(\hat{V}_{\mathrm{LIPS}}(\pi;\mathcal{D}))^2
$$

$$
= \mathbb{E}_{\mathcal{D}}[(\hat{V}_{\mathrm{IPS}}(\pi;\mathcal{D}))^2] - \mathbb{E}_{\mathcal{D}}[(\hat{V}_{\mathrm{LIPS}}(\pi;\mathcal{D}))^2] - ((\mathbb{E}_{\mathcal{D}}[\hat{V}_{\mathrm{IPS}}(\pi;\mathcal{D})])^2 - \mathbb{E}_{\mathcal{D}}[\hat{V}_{\mathrm{LIPS}}(\pi;\mathcal{D})])^2)
$$

$$
- n\mathrm{Bias}(\hat{V}_{\mathrm{LIPS}}(\pi;\mathcal{D}))^2
$$

$$
= \mathbb{E}_{p(\boldsymbol{x})\pi_0(s\,|\,\boldsymbol{x})p_\theta(z\,|\,\boldsymbol{x},s)p(r\,|\,\boldsymbol{x},s)}[(w(\boldsymbol{x},s)r)^2 - (w_\theta(\boldsymbol{x},z)r)^2]
$$

$$
- ((V(\pi))^2 - (V(\pi) + \mathrm{Bias}(\hat{V}_{\mathrm{LIPS}}(\pi;\mathcal{D})))^2) - n\mathrm{Bias}(\hat{V}_{\mathrm{LIPS}}(\pi;\mathcal{D}))^2
$$

$$
= \mathbb{E}_{p(\boldsymbol{x})\pi_0(s\,|\,\boldsymbol{x})p_\theta(z\,|\,\boldsymbol{x},s)p(r\,|\,\boldsymbol{x},s)}[((w(\boldsymbol{x},s))^2 - (w_\theta(\boldsymbol{x},z))^2) \cdot r^2]
$$

$$
+ 2V(\pi)\mathrm{Bias}(\hat{V}_{\mathrm{LIPS}}(\pi;\mathcal{D})) + \mathrm{Bias}(\hat{V}_{\mathrm{LIPS}}(\pi;\mathcal{D}))^2 - n\mathrm{Bias}(\hat{V}_{\mathrm{LIPS}}(\pi;\mathcal{D}))^2
$$

$$
= \mathbb{E}_{p(\boldsymbol{x})\pi_0(s\,|\,\boldsymbol{x})p_\theta(z\,|\,\boldsymbol{x},s)}[(w(\boldsymbol{x},s)^2 - w_\theta(\boldsymbol{x},z)^2) \cdot \mathbb{E}_{p(r\,|\,\boldsymbol{x},s)}[r^2]]
$$

$$
+ 2V(\pi)\mathrm{Bias}(\hat{V}_{\mathrm{LIPS}}(\pi;\mathcal{D})) + (1-n)\mathrm{Bias}(\hat{V}_{\mathrm{LIPS}}(\pi;\mathcal{D}))^2
$$

Then, the first term is further decomposed to variance and co-variance as follows.

$$
\mathbb{E}_{p(\boldsymbol{x})\pi_0(s\,|\,\boldsymbol{x})p_\theta(z\,|\,\boldsymbol{x},s)}[(w(\boldsymbol{x},s)^2 - w_\theta(\boldsymbol{x},z)^2) \cdot \mathbb{E}_{p(r\,|\,\boldsymbol{x},s)}[r^2]]
$$

$$
= \mathbb{E}_{p(\boldsymbol{x})\pi_0(s\,|\,\boldsymbol{x})p_\theta(z\,|\,\boldsymbol{x},s)}[(w(\boldsymbol{x},s)^2 - w_\theta(\boldsymbol{x},z)^2) \cdot \mathbb{E}_{p(r\,|\,\boldsymbol{x},s)}[r^2]]
$$

$$
= \mathbb{E}_{p(\boldsymbol{x})}\big[\sum_{s\in\mathcal{S}} \pi_0(s\,|\,\boldsymbol{x}) \sum_{z\in\mathcal{Z}} p_\theta(z\,|\,\boldsymbol{x},s)(w(\boldsymbol{x},s)^2 - w_\theta(\boldsymbol{x},z)^2) \cdot \mathbb{E}_{p(r\,|\,\boldsymbol{x},s)}[r^2]\big]
$$

$$
= \mathbb{E}_{p(\boldsymbol{x})}\big[\sum_{s\in\mathcal{S}} \pi_0(s\,|\,\boldsymbol{x}) \sum_{z\in\mathcal{Z}} \frac{p_\theta(z\,|\,\boldsymbol{x},\pi_0)p_\theta(s\,|\,\boldsymbol{x},z,\pi_0)}{\pi_0(s\,|\,\boldsymbol{x})}(w(\boldsymbol{x},s)^2 - w_\theta(\boldsymbol{x},z)^2) \cdot \mathbb{E}_{p(r\,|\,\boldsymbol{x},s)}[r^2]\big]
$$

$$
= \mathbb{E}_{p(\boldsymbol{x})}\big[\sum_{z\in\mathcal{Z}} p_\theta(z\,|\,\boldsymbol{x},\pi_0) \sum_{s\in\mathcal{S}} p_\theta(s\,|\,\boldsymbol{x},z,\pi_0)(w(\boldsymbol{x},s)^2 - w_\theta(\boldsymbol{x},z)^2) \cdot \mathbb{E}_{p(r\,|\,\boldsymbol{x},s)}[r^2]\big]
$$

$$
= \mathbb{E}_{p(\boldsymbol{x})p_\theta(z\,|\,\boldsymbol{x},\pi_0)p_\theta(s\,|\,\boldsymbol{x},z,\pi_0)}[(w(\boldsymbol{x},s)^2 - w_\theta(\boldsymbol{x},z)^2) \cdot \mathbb{E}_{p(r\,|\,\boldsymbol{x},s)}[r^2]]
$$

$$
= \mathbb{E}_{p(\boldsymbol{x})p_\theta(z\,|\,\boldsymbol{x},\pi_0)p_\theta(s\,|\,\boldsymbol{x},z,\pi_0)}[(w(\boldsymbol{x},s)^2 - (\mathbb{E}_{p_\theta(s\,|\,\boldsymbol{x},z,\pi_0)}[w(\boldsymbol{x},s)])^2) \cdot \mathbb{E}_{p(r\,|\,\boldsymbol{x},s)}[r^2]]
$$

$$
= \mathbb{E}_{p(\boldsymbol{x})p_\theta(z\,|\,\boldsymbol{x},\pi_0)p_\theta(s\,|\,\boldsymbol{x},z,\pi_0)}[(w(\boldsymbol{x},s)^2 - (\mathbb{E}_{p_\theta(s\,|\,\boldsymbol{x},z,\pi_0)}[w(\boldsymbol{x},s)])^2)
$$

$$
\cdot (\mathbb{E}_{p_\theta(s\,|\,\boldsymbol{x},z,\pi_0)}[\mathbb{E}_{p(r\,|\,\boldsymbol{x},s)}[r^2]] + (\mathbb{E}_{p(r\,|\,\boldsymbol{x},s)}[r^2] - \mathbb{E}_{p_\theta(s\,|\,\boldsymbol{x},z,\pi_0)}[\mathbb{E}_{p(r\,|\,\boldsymbol{x},s)}[r^2]]))]
$$

$$
= \mathbb{E}_{p(\boldsymbol{x})p_\theta(z\,|\,\boldsymbol{x},\pi_0)p_\theta(s\,|\,\boldsymbol{x},z,\pi_0)}[(w(\boldsymbol{x},s)^2 - (\mathbb{E}_{p_\theta(s\,|\,\boldsymbol{x},z,\pi_0)}[w(\boldsymbol{x},s)])^2) \cdot \mathbb{E}_{p_\theta(s\,|\,\boldsymbol{x},z,\pi_0)}[\mathbb{E}_{p(r\,|\,\boldsymbol{x},s)}[r^2]]]
$$

$$
+ \mathbb{E}_{p(\boldsymbol{x})p_\theta(z\,|\,\boldsymbol{x},\pi_0)p_\theta(s\,|\,\boldsymbol{x},z,\pi_0)}[(w(\boldsymbol{x},s)^2 - (\mathbb{E}_{p_\theta(s\,|\,\boldsymbol{x},z,\pi_0)}[w(\boldsymbol{x},s)])^2)
$$

$$
\cdot (\mathbb{E}_{p(r\,|\,\boldsymbol{x},s)}[r^2] - \mathbb{E}_{p_\theta(s\,|\,\boldsymbol{x},z,\pi_0)}[\mathbb{E}_{p(r\,|\,\boldsymbol{x},s)}[r^2]])]
$$

$$
= \mathbb{E}_{p(\boldsymbol{x})p_\theta(z\,|\,\boldsymbol{x},\pi_0)}\big[\mathbb{V}_{p_\theta(s\,|\,\boldsymbol{x},z,\pi_0)}(w(\boldsymbol{x},s)) \cdot \mathbb{E}_{p_\theta(s\,|\,\boldsymbol{x},z,\pi_0)}\big[\mathbb{E}_{p(r\,|\,\boldsymbol{x},s)}[r^2]\big]\big]
$$

$$
+ \mathbb{E}_{p(\boldsymbol{x})p_\theta(z\,|\,\boldsymbol{x},\pi_0)}\big[\mathrm{Cov}_{p_\theta(s\,|\,\boldsymbol{x},z,\pi_0)}\big(w(\boldsymbol{x},s)^2,\, \mathbb{E}_{p(r\,|\,\boldsymbol{x},s)}[r^2]\big)\big]
$$