# OpenReview forum: "Off-Policy Evaluation of Slate Bandit Policies via Optimizing Abstraction"
_ACM.org/TheWebConf/2024/Conference — TheWebConf24_

### Official Review · Reviewer_krDV · 2023-11-02

**Novelty:** 5
**Technical Quality:** 4

**Review:**

Off-Policy Evaluation of Slate Bandit Policies via Optimizing Abstraction

The core idea in this paper is to replace the usual IPS ratio of pi(s|x)/pi0(s|x) where s is a large combinatorial slate with pi(z|x)/pi0(z|x) where z is a low dimension projection of s.  In cases where z has lower cardinality than s, this results in an estimator with lower variance (which is typically horrific if s is a slate of even modest size).

The authors present theorems on how these projections impact the bias and the variance of the estimators.

They then propose a loss that allows optimization of the mapping from s to z a method that they call LIPS.  Promising experimental results of the method are presented.

I have one primary concern about the method.  The algorithm proposed attempts to reduce variance by creating a ratio pi/pi0 that is close to one - by grouping actions together.  Unless I missed it there is nothing to cause similar actions to be grouped together in this process - and therefore important information regarding the similarities between actions is ignored.

Comments:

In Equation 1 - Independent policies on slates are usually not desirable as they allow the slate to contain repeats of items.

S. Li, Y. Abbasi-Yadkori, B. Kveton, S. is worth citing with respect to Equation 2.

A further important remark is that marginalizing both pi and pi0 is usually computationally infeasible unless an assumption like independence is made – and this is undesirable as it allows slates to contain repeats of the same recommendation.

**Questions:**

It would seem that the estimation will be improved if the mapping from s to z places similar actions/slates in the same cluster.  However the optimization problem 9 seems indifferent to the items.  Would it be possible to modify the algorithm so it can also use information about similarities between actions?

**Reviewer Confidence:**

2: The reviewer is willing to defend the evaluation, but it is likely that the reviewer did not understand parts of the paper

**Scope:**

4: The work is relevant to the Web and to the track, and is of broad interest to the community

---

### Official Review · Reviewer_sTFy · 2023-11-16

**Novelty:** 6
**Technical Quality:** 3

**Review:**

Summary: This paper propose a new off-policy evaluation method of slate bandit policies by state abstraction. The paper proposes the LIPS estimator, which uses the abstracted state to get an unbiased estimator. The paper also gives an optimization method to learn the abstraction function. Experiments show that LIPS outperforms PI, IPS and MIPS in real datasets.

Strength: The paper proposes a novel unbiased estimator for slate bandits policy evaluations. The paper provides some theoretical analysis. The paper also propose a learning method to optimize the abstraction function.

Weakness: The motivation that uses a sampling policy to sample z from the latent x is unclear, and it is quite complex. The paper does not justify the convergence of Loss(9). How to ensure that the abstraction function satisfies the sufficient slate abstraction condition when (9) is zero? The paper does not cite and compare the DR method for ranking(Doubly Robust Off-Policy Evaluation for Ranking Policies under the Cascade Behavior Model, WSDM 2022).

**Questions:**

See the above section.

**Reviewer Confidence:**

3: The reviewer is confident but not certain that the evaluation is correct

**Scope:**

4: The work is relevant to the Web and to the track, and is of broad interest to the community

---

### Official Review · Reviewer_Vzwv · 2023-11-22

**Novelty:** 5
**Technical Quality:** 6

**Review:**

This paper deals with the problem of off-policy evaluation for slate bandit policies. Specifically, the goal is to estimate the performance of a strategy for selecting a collection of slot-dependent actions without explicity testing it, but by using a historical data set. While there has been a significant amount of work on off-policy estimation in general, these approaches are generally subject to high variance when the action space is large, such as in slate bandit problems. Specific to the slate bandit problem, there has been relatively less work to develop specialized solutions. Those that do primarily rely on somewhat restrictive linear reward assumptions. This work devices an off-policy estimator for slate bandits, that does not make linear reward assumptions, and examines its theoretical guarantees and evaluates the empirical comparison to alternatives. The approach is based on the idea of having a slate abstraction function that maps slates to a a latent space. The estimator is then effectively IPS but using the latent importannce weights. The theoretical guarantees show that under a fairly restriction assumption (sufficient abstraction) that the estimator is unbiased. The estimator is show to be variance reducing compared to the usual IPS method. Also, the bias is analyzed dependent on the slate abstraction, and this set of analysis is used to devise an optimization problem to minimize the bias and variance of the estimator with a choice of slate abstraction.

The paper is well-motivated and easy to follow. I think this paper meets the bar for acceptance. The overall objective (off-policy evaluation for slates without linearity assumptions) is important, and the estimation approach is intuitive. The corresponding theoretical guarantees (bias and variance) are comprehensive for the estimator, and these are used to make the estimator more effective since the analysis lends to a method for creating the slate abstraction. The empirical results drive home the main points of the paper. In terms of some potential limiting factors, this paper is a somewhat specialized topic and it is novel but not extremely so; specifcally the approach is not all that different from the MIPS estimator, but does have the advantage of not needing to be given the action embeddings ahead of time.


Pros
- Well-movtivated practical problem
- Fairly comprehensive theoretical and empirical story

Cons
- It could be argued that the contributions are somewhat incremental compared to the existing literature.

**Questions:**

To be clear, I think it is a good paper, but if the authors feel I am understanding the novelty, please let me know what I have missed.

**Reviewer Confidence:**

3: The reviewer is confident but not certain that the evaluation is correct

**Scope:**

3: The work is somewhat relevant to the Web and to the track, and is of narrow interest to a sub-community

---

### Official Review · Reviewer_ZXMZ · 2023-12-01

**Novelty:** 5
**Technical Quality:** 4

**Review:**

Strengths:
- Good explanation and theoretical justification for using LIPS over existing methods for Off-Policy Evaluation (OPE), such as MIPS, PI, IPS, DM, and DR.
- Real-world motivation seems useful.

Weaknesses:
- The diagrams exclusively look at MSE but considering the motivation for LIPS is to improve both bias and variance, I think it would have been useful to separate the MSE into the bias and variance.  This would showcase these claimed advantages of LIPS more clearly, i.e., do both bias and variance actually improve as claimed, or only one?
- Some diagrams have low error bars despite having large jumps (e.g., LIPS performance in Figure 2 (middle)), which may indicate improper randomization in the experimentation.  The reasons for these high-confidence large jumps was not discussed in the paper.
- The semi-synthetic data generation using over extreme classification datasets is not clearly reflective of the same type of problems provided as motivation for slate evaluation for recommender and advertising systems.

This paper proposes a new estimation technique for Off-Policy Evaluation (OPE) called Latent Inverse Propensity Scoring (LIPS). The paper claims that limitations of the existing techniques are as follows: IPS has higher variance (but no bias), Direct Method (DM) has low variance but high bias, PseudoInverse (PI) is unbiased but only under an assumption of linearity and can have a high variance when there are many unique sub-actions. The specified goal for the LIPS estimator is to improve the bias and variance over other estimators and provides a way to control the balance between the bias and variance using a hyperparameter \beta. Specifically, the paper proposes to do this by not making the linearity assumption and by finding a latent slate space abstraction that does not affect the rewards, which would theoretically lead to better performance on real world data. A data-driven optimization procedure is proposed to obtain an appropriate insufficient slate abstraction, which is a crucial component of LIPS.

There are however, 2 main concerns I have with the paper:

(1) Since the paper claims that the LIPS estimator improves upon the limitations of other estimators on real-world data and provides slate evaluation for recommender and advertising systems as the motivation, it would be important for experiments to have been conducted on real-world data that reflect these types of problems rather than on semi-synthetic extreme classification datasets. For example, the (statistical) relationships between data labels and documents may not be reflective of the same type of relationships between components of an advertisement and advertising campaign data.

(2) As particularly visible in Figure 2 (middle), there are erratic jumps up and down in the MSE vs. slate size, however the confidence is quite high (low error bars), which is not explained or justified in the paper and may be an indication of improper randomization in the algorithms or evaluation.

**Questions:**

(1) Is there a justification or explanation for the zig-zag behaviour in Figure 2 (middle) and why the confidence is high?

(2) Is there clear justification for the datasets used and how they relate to real world problems in the domain of slate evaluation for recommender and advertising systems?

**Reviewer Confidence:**

3: The reviewer is confident but not certain that the evaluation is correct

**Scope:**

3: The work is somewhat relevant to the Web and to the track, and is of narrow interest to a sub-community

---

### Decision · Program_Chairs · 2024-01-22

**Decision:**

Accept

**Comment:**

The proposes a novel methodology for off-policy evaluation of slate bandits that relaxes linearity assumptions in previous work.

 Reviewers raise a range of technical and experimental concerns, which the authors have carefully responded to.

 I do want to follow up on two points of reviewer-author discussion with ZXMZ:

 1. Reviewer ZXMZ's has concerns that the paper motivates with recommendation and advertising systems, but that the experiments are on a semi-synthetic extreme classification dataset. The authors' rebuttal notes that there are no relevant public datasets for "datasets for slate bandits with slot actions" and effectively that their experimental evaluation is consistent with previous publications. While the authors' points are taken here, it does seem at a minimum that they could have tried to set up a simulation of a recommendation or advertising system as a (semi-)synthetic evaluation much more closely targeted to their motivating use cases for this work.

 2. Reviewer ZXMZ also has concerns about Figure 2(middle). As for the reviewer, I am also concerned that the performance zig-zags by an order of magnitude between slate sizes 6, 8, and 10 and the confidence intervals indicate that we should be confident about this performance. But shouldn't slate size 8 be able to capture the performance of slate size 6? Certainly slate size 10 can do this, as shown. The authors explain that there is actually little absolute performance difference since this is a log scale. But I don't think that directly addresses the reviewer's concern as I see it... it's not the difference here that is the concern, but the *high confidence* in these order of magnitude differences. Overall, I am concerned that there may be some lack of randomization in the experiments that is providing an overestimated confidence for slate size 8. I have no way of confirming this, so it is only a conjecture, but at the very least, this anomalous behavior needs to be clearly explained in the paper since it does clearly stand out in these plots.

 Overall, the paper makes a solid technical contribution though there a few minor details discussed with reviewers where revision/clarification would be useful (which is expected for any paper submission). Perhaps more importantly and the reason I cannot push it to a clear accept rating is that it falls short in terms of directly evaluating it's motivating use cases.